# Robust Federated Learning Against Adaptive Compression

Wenjing Yan [1]    Xiangyu Zhong [1]    Ying-Jun Angela Zhang [1]

## Abstract

This paper studies the intertwined challenges of hyperparameter tuning and communication efficiency in federated learning (FL). Although recent parameter-free FL methods such as PAdaMFed remove the need to tune problem-dependent parameters, their substantial communication overhead remains a major barrier to practical deployment. To address this limitation, we propose ParFreFL, a communication-efficient parameter-free FL algorithm that reduces the communication cost of PAdaMFed by half while preserving its parameter-free nature. We further develop a compressed variant, ComParFreFL, which combines momentum increments and error feedback into a single transmitted quantity, thereby accommodating biased compression without increasing communication cost. Notably, ComParFreFL is independent of the compression ratio, offering, to the best of our knowledge, the first such robustness guarantee in compressed FL. Theoretically, our methods accommodate arbitrary data heterogeneity and partial client participation, and achieve linear speedup with respect to both the number of local updates and the number of participating clients. Extensive experiments show that the proposed algorithms match or outperform carefully tuned baselines while substantially reducing communication overhead, making parameter-free FL more practical for dynamic and resource-constrained environments.

## 1. Introduction

Parameter-free optimization has emerged as a promising direction in machine learning, addressing one of the central challenges in modern optimization: selecting algorithmic hyperparameters (Khaled & Jin, 2024; Attia & Koren, 2024). Conventional optimization methods often depend heavily on carefully chosen learning rates, momentum coefficients, regularization parameters, and other problem-dependent quantities. Selecting these parameters typically requires domain expertise, substantial computational resources, and extensive trial and error. This process is not only time-consuming but can also lead to suboptimal performance, especially when data distributions are diverse or evolve over time. To mitigate this burden, "tuning-free" or "problem-parameter-free" optimization strategies have attracted increasing attention. By adapting to intrinsic problem properties, these methods provide more streamlined and reliable optimization procedures (Kreisler et al., 2024; Yang et al., 2023). Recent progress has demonstrated their potential across a range of settings, including nonconvex optimization (Li et al., 2024), min-max problems (Huang et al., 2024b), bilevel optimization (Yang et al., 2024), and Riemannian optimization (Dodd et al., 2024).

Federated learning (FL) (McMahan et al., 2017) provides a particularly compelling and challenging testbed for parameter-free optimization. As a distributed learning paradigm, FL enables decentralized clients to collaboratively train a shared model while keeping data local. However, clients often differ substantially in data distributions, computational resources, and network conditions, making conventional hyperparameter tuning especially difficult (Karimireddy et al., 2020b; Cheng et al., 2024). In compressed FL, the dependence on problem-specific parameters—such as learning rates, momentum coefficients, and compression ratios—further compounds this difficulty and can hinder scalability, adaptability, and deployment efficiency (Huang et al., 2024a; Li & Li, 2023; Haddadpour et al., 2021). Parameter-free optimization therefore offers a natural route toward more practical FL systems in heterogeneous real-world environments.

Prior work has explored adaptive mechanisms to reduce the tuning burden in FL. FedAdam and FedYogi (Reddi et al., 2020) apply server-side adaptive optimization, but require additional optimizer states. FedOpt (Asad et al., 2020) provides a general framework for incorporating adaptive server optimizers. Local adaptive methods, such as FedLion (Tang & Chang, 2024), maintain client-side adaptive states to reduce communication overhead while

---

[1]Department of Information Engineering, The Chinese University of Hong Kong, Hong Kong SAR;. Correspondence to: Xiangyu Zhong <xyzhong@ie.cuhk.edu.hk>.

*Proceedings of the 43rd International Conference on Machine Learning*, Seoul, South Korea. PMLR 306, 2026. Copyright 2026 by the author(s).

retaining the benefits of adaptivity. Recent hybrid methods have also been studied: FedAGM (Ba et al., 2024) combines global momentum with local adaptivity, whereas FAdamGC (Chen et al., 2025) uses adaptive gradient scaling to address non-IID data. Despite their strong empirical performance, these methods still rely on problem-specific hyperparameters that can substantially affect both convergence and generalization.

Recent progress in parameter-free FL has led to PAdaMFed (Yan et al., 2024), a pioneering method that eliminates the need to manually tune problem-specific hyperparameters. Despite its theoretical appeal, PAdaMFed faces a practical limitation: substantial communication overhead. In particular, each communication round requires clients and the server to exchange two model-sized variables in both upload and download phases, doubling the communication burden compared with conventional FedAvg-based approaches. This limitation highlights the need for communication-efficient parameter-free FL. A natural remedy is compression, a widely used technique for reducing transmission volume. However, incorporating compression into the PAdaMFed framework is nontrivial. Simultaneously compressing two variables introduces coupled compression errors, which complicate the convergence analysis and can destabilize the learning process.

### 1.1. Main Contributions

In this paper, we address these challenges by proposing a communication-efficient **Par**ameter-**Fre**e **FL** algorithm, **ParFreFL**, which halves the communication cost of PAdaMFed while preserving its parameter-free property. Building on this foundation, we further introduce **ComParFreFL**, a compressed variant that unifies the momentum increment and error feedback into a single quantity, effectively handling biased compression while retaining minimal communication cost. Our main contributions are summarized below.

- We propose ParFreFL, a communication-efficient parameter-free FL algorithm that halves the communication burden of PAdaMFed-based approaches by transmitting only one model-sized quantity per update. All learning rates in ParFreFL are explicitly determined solely by predefined system constants, eliminating the need for hyperparameter tuning. Our theoretical analysis shows that ParFreFL retains state-of-the-art convergence guarantees while accommodating arbitrary data heterogeneity and partial client participation. Notably, ParFreFL improves the communication complexity of PAdaMFed, requiring $\mathcal{O}(\epsilon^{-2})$ communication rounds to achieve an accuracy of $\mathbb{E}\|f(\boldsymbol{\theta})\| \leq \epsilon$, whereas PAdaMFed requires $\mathcal{O}(\epsilon^{-3})$ rounds.

- We develop ComParFreFL, a compressed variant that further improves the communication efficiency of parameter-free FL. ComParFreFL handles biased compression while preserving the minimal communication cost of ParFreFL. This is achieved through a compression mechanism that unifies the momentum increment and error feedback into a single quantity, allowing compression bias to be controlled without sacrificing the parameter-free property. Our theoretical analysis shows that, for sufficiently large $T$, ComParFreFL achieves the same convergence order as the full-transmission algorithm ParFreFL. Moreover, ComParFreFL operates independently of the compression ratio, enabling clients to dynamically adjust their communication schemes in response to changing network conditions.

- We conduct comprehensive empirical evaluations on deep learning tasks with real-world datasets. The results show that our methods consistently outperform state-of-the-art compressed FL algorithms in runtime efficiency, even when the competing algorithms are carefully tuned. These gains stem from the acceleration provided by our momentum mechanism and the per-step adaptation of effective stepsizes.

## 2. Communication-Efficient Parameter-Free Federated Learning

### 2.1. Problem Formulation

We consider the standard FL setting where a central server coordinates model training across $N$ distributed clients. The global objective is to minimize:

$$\min_{\boldsymbol{\theta} \in \mathbb{R}^d} f(\boldsymbol{\theta}) := \frac{1}{N} \sum_{i=1}^{N} f_i(\boldsymbol{\theta})$$
$$\text{s.t. } f_i(\boldsymbol{\theta}) := \mathbb{E}_{\boldsymbol{\xi}_i \sim \mathcal{D}_i} \left[ F\left(\boldsymbol{\theta}; \boldsymbol{\xi}_i\right) \right].$$

Here, $\boldsymbol{\theta} \in \mathbb{R}^d$ represents the global model parameters, and $f(\boldsymbol{\theta})$ denotes the global objective function averaged across all clients. Each client $i \in [N]$ has a local objective function $f_i(\boldsymbol{\theta})$, defined as the expected loss over its local data distribution $\mathcal{D}_i$. The local loss function $F(\boldsymbol{\theta}; \boldsymbol{\xi}_i)$ computes the model's performance on data sample $\boldsymbol{\xi}_i$ drawn from distribution $\mathcal{D}_i$.

The key challenge in FL lies in solving this optimization problem in a distributed manner while addressing the following constraints:

1. **Data Heterogeneity:** The local data distributions $\mathcal{D}_i$ can vary significantly betweenn clients, resulting in non-i.i.d. data.

**Algorithm 1** ParFreFL: Parameter-Free Federated Learning

---

1: **Require:** Initial model $\boldsymbol{\theta}^0$, $\boldsymbol{m}_i^{-1} = \frac{1}{K}\sum_{k=0}^{K-1}\nabla F\left(\boldsymbol{\theta}^0;\boldsymbol{\xi}_i^{-1,k}\right)$ and $\boldsymbol{c}_i^{-1} = \boldsymbol{m}_i^{-1}$ for any $i$, $\boldsymbol{c}^{-1} = \frac{1}{N}\sum_i \boldsymbol{c}_i^{-1}$, learning rates $\eta$ and $\gamma$, and momentum parameter $\beta$

2: **for** $t = 0, \cdots, T-1$ **do**

3:    **Central Server:** Uniformly sample clients $\mathcal{S}_t \subseteq \{1, \cdots, N\}$ with $|\mathcal{S}_t| = S$

4:    **for** each client $i \in \mathcal{S}_t$ in parallel **do**

5:       Receive $\boldsymbol{\theta}^t$ and initialize $\boldsymbol{\theta}_i^{t,0} = \boldsymbol{\theta}^t$

6:       **for** $k = 1, \cdots, K$ **do**

7:          Compute
$$\boldsymbol{m}_i^{t,k} = (1-\beta)\boldsymbol{m}_i^{t-1} + \beta\nabla F\left(\boldsymbol{\theta}_i^{t,k};\boldsymbol{\xi}_i^{t,k}\right)$$

8:          Update local model $\boldsymbol{\theta}_i^{t,k+1} = \boldsymbol{\theta}_i^{t,k} - \eta\frac{\boldsymbol{m}_i^{t,k}}{\|\boldsymbol{m}_i^{t,k}\|}$

9:       **end for**

10:      Upload $\boldsymbol{m}_i^t = \frac{1}{K}\sum_k \boldsymbol{m}_i^{t,k}$
       ($\boldsymbol{m}_i^t = \boldsymbol{m}_i^{t-1}$ for $i \notin \mathcal{S}_t$ )

11:    **end for**
   **Central server:**

12:    Set $\boldsymbol{c}_i^t = \boldsymbol{m}_i^t$ if $i \in \mathcal{S}_t$, and $\boldsymbol{c}_i^t = \boldsymbol{c}_i^{t-1}$ otherwise

13:    Aggregate control variate
$$\boldsymbol{c}^t = \boldsymbol{c}^{t-1} + \frac{1}{N}\sum_{i\in\mathcal{S}_t}\left(\boldsymbol{c}_i^t - \boldsymbol{c}_i^{t-1}\right)$$

14:    Compute $\boldsymbol{g}^t = \frac{1}{S}\sum_{i\in\mathcal{S}_t}\left(\boldsymbol{c}_i^t - \boldsymbol{c}_i^{t-1}\right) + \boldsymbol{c}^{t-1}$

15:    Update global model $\boldsymbol{\theta}^{t+1} = \boldsymbol{\theta}^t - \gamma\frac{\boldsymbol{g}^t}{\|\boldsymbol{g}^t\|}$

16:    Download $\boldsymbol{\theta}^{t+1}$ to all clients

17: **end for**

---

2. **Communication Efficiency:** Limited communication resources between clients and the central server necessitate techniques to reduce communication overhead.

3. **Algorithm Tuning:** Proper calibration of algorithmic parameters (learning rates, momentum coefficients, and compression rates) is critical to ensure convergence, particularly in dynamic and heterogeneous environments.

This paper presents a unified FL framework that addresses these challenges, enabling large-scale, privacy-preserving model training across distributed and heterogeneous datasets.

### 2.2. Development of ParFreFL

In this subsection, we propose **ParFreFL**, a communication-efficient, parameter-free federated learning algorithm, as outlined in Algorithm 1. ParFreFL is designed to address three central challenges in federated settings: hyperparameter tuning, data heterogeneity, and partial client participation. Although our approach builds on standard ingredients such as momentum, control variates, and compression, its design is far from a straightforward combination of these components. The main technical novelty lies in the development and analysis of a parameter-free FL framework that simultaneously reduces communication overhead and accommodates biased compression, all without requiring compression-dependent tuning rules.

**Eliminating Hyperparameter Tuning.** ParFreFL eliminates the need for manual tuning of algorithmic hyperparameters, a significant bottleneck in traditional FL approaches. In standard gradient-descent-based optimization, learning rates must be sufficiently small to accommodate gradient magnitudes, determined by the smoothness constant $L$. Excessive stepsizes can cause overshooting and instability. In the FL setting, algorithm performance typically depends on additional problem-specific parameters, further complicating hyperparameter tuning. For instance, MIME (Karimireddy et al., 2020a) requires a complex learning rate configuration dependent on the smoothness constant $L$, initial optimality gap $\Delta$, stochastic gradient variance $\sigma^2$, and heterogeneity bound $\sigma_h^2$. Similarly, FedSPS (Sohom Mukherjee, 2024) requires prior knowledge of local loss function lower bounds and $L$, while SCAFFOLD-M's (Cheng et al., 2024) stepsizes depend on multiple parameters including $L$, $\Delta$, $\sigma^2$, and $\frac{1}{N}\sum_i\|\nabla f(\boldsymbol{\theta}_0)\|^2$. Estimating these parameters in federated settings is highly challenging and typically requires extensive trial-and-error processes.

ParFreFL addresses these challenges through gradient normalization at both local and global levels (Lines 7 and 14, Algorithm 1). This normalization ensures stepsize independent of gradient magnitude, eliminating reliance on the smoothness constant $L$ and other problem-specific parameters. However, gradient normalization alone is insufficient for stochastic optimization as it discards magnitude information. This limitation can lead to suboptimal updates when randomness in sample selection causes descent directions to be dominated by gradients from larger mini-batches that may not align with the true descent direction (Yang et al., 2023). To overcome this issue, ParFreFL incorporates a momentum term $\boldsymbol{m}_i^t$ in the local update (Line 6, Algorithm 1). This momentum accumulates historical gradient information, smoothing stochastic gradient noise and promoting more consistent descent directions. By combining gradient normalization with momentum, ParFreFL eliminates hyperparameter tuning while ensuring stability and robust convergence across diverse settings.

**Handling Arbitrary Heterogeneous Data.** ParFreFL demonstrates robust performance with arbitrarily non-i.i.d. data across clients, a significant challenge in FL. We ad-

dress this through a novel control variate design that incorporates both stochastic gradients and momentum. The local control variate is defined as:

$$\boldsymbol{c}_i^t = \boldsymbol{m}_i^t = (1-\beta)\boldsymbol{m}_i^{t-1} + \tfrac{\beta}{K}\sum_k \nabla F\left(\boldsymbol{\theta}_i^{t,k}; \boldsymbol{\xi}_i^{t,k}\right).$$

The global descent direction is then given by:

$$\boldsymbol{g}^t = \tfrac{1}{S}\sum_{i\in\mathcal{S}_t}\big((1-\beta)\boldsymbol{m}_i^{t-1}$$
$$+ \tfrac{\beta}{K}\sum_k \nabla F\left(\boldsymbol{\theta}_i^{t,k}; \boldsymbol{\xi}_i^{t,k}\right) - \boldsymbol{c}_i^{t-1}\big) + \boldsymbol{c}^{t-1}.$$

The aggregation rule for the global control variate (Line 12, Algorithm 1) ensures that $\boldsymbol{c}^t = \frac{1}{N}\sum_i \boldsymbol{c}_i^t$ always holds. Let

$$\boldsymbol{m}^t := \tfrac{1}{N}\sum_i \boldsymbol{m}_i^t$$
$$= \tfrac{1}{N}\sum_i \left((1-\beta)\boldsymbol{m}_i^{t-1} + \tfrac{\beta}{K}\sum_k \nabla F\left(\boldsymbol{\theta}_i^{t,k}; \boldsymbol{\xi}_i^{t,k}\right)\right)$$

represent the global descent vector under full client participation. Since $\boldsymbol{c}_i^t = \boldsymbol{m}_i^t$ holds for all $i$ and $t$, we can bound the gradient deviation due to partial client participation by:

$$\mathbb{E}\left\|\boldsymbol{m}^t - \boldsymbol{g}^t\right\|^2$$
$$\leq \tfrac{\beta^2}{SN}\sum_i \mathbb{E}\left\|\boldsymbol{m}_i^{t-1} - \tfrac{1}{K}\sum_k \nabla F\left(\boldsymbol{\theta}_i^{t,k}; \boldsymbol{\xi}_i^{t,k}\right)\right\|^2,$$

which depends solely on discrepancies between stochastic gradients and momentum at each client, eliminating the need for bounds on cross-client data heterogeneity.

**Accommodating Partial Participation.** ParFreFL efficiently handles partial client participation, a common scenario in FL where only a subset of clients $\mathcal{S}_t$ participate in each communication round. This is achieved by incorporating control variate discrepancies $\boldsymbol{c}_i^{t-1} - \boldsymbol{c}^{t-1}$ (Line 13, Algorithm 1) into global updates, following the SCAFFOLD design (Karimireddy et al., 2020b).

### 2.3. Comparisons with PAdaMFed

PAdaMFed requires the transmission of two model-sized parameters during both upload and download processes. The uplink transmissions include $\boldsymbol{\theta}_i^{t,K}$ and $\boldsymbol{c}_i^t := \frac{1}{K}\sum_k \nabla F(\boldsymbol{\theta}_i^{t,k}; \boldsymbol{\xi}_i^{t,k})$, where $\boldsymbol{\theta}_i^{t,K}$ is used to compute accumulated local gradients $\frac{1}{S}\sum_{i\in\mathcal{S}_t}(\boldsymbol{\theta}^t - \boldsymbol{\theta}_i^{t,K})$ that determine the global descent direction. Additionally, $\boldsymbol{c}_i^t$ updates the global control variate $\boldsymbol{c}^t$ and global momentum $\boldsymbol{g}^t$. For downlink, the updated global model $\boldsymbol{\theta}^{t+1}$ and combined global control variate and momentum $\beta\boldsymbol{c}^t + (1-\beta)\boldsymbol{g}^t$ are transmitted to clients.

In contrast, ParFreFL reduces communication overhead by half through two key modifications:

**1) Distinct Descent Direction.** In **ParFreFL**, the global model is updated along the global momentum vector $\boldsymbol{g}^t$,

which is computed from the control variates $\boldsymbol{c}_i^t, i \in \mathcal{S}_t$. Consequently, the server no longer requires access to the local models $\boldsymbol{\theta}_i^{t,K}$, thereby reducing uplink communication costs by half. Notably, the direction of $\boldsymbol{g}^t$ differs from the accumulated local descent directions employed in PAdaMFed, resulting in distinct descent directions for local and global updates.

**2) Client-Specific Momentum.** **ParFreFL** implements client-specific momentum that accumulates only local historical gradient directions. In contrast, PAdaMFed employs globally aggregated momentum that requires server-to-client broadcast, incurring additional communication overhead.

These modifications, however, introduce additional challenges for theoretical analysis. The discrepancies between global and local descent directions and the use of client-specific momentum must be carefully managed to preserve the convergence properties of PAdaMFed.

### 2.4. Theoretical Results

The convergence analysis of ParFreFL is based on the following assumptions:

**Assumption 1** (*L*-Smoothness). *For any $i \in [N]$, the local loss function $f_i(\cdot)$ is L-smooth that*

$$\left\|\nabla f_i(\boldsymbol{\theta}) - \nabla f_i(\boldsymbol{\theta}')\right\| \leq L\|\boldsymbol{\theta} - \boldsymbol{\theta}'\|, \quad \forall \boldsymbol{\theta}, \boldsymbol{\theta}' \in \mathbb{R}^d.$$

**Assumption 2** (Stochastic Gradient). *The stochastic gradients are unbiased estimators with bounded variance $\sigma^2$, such that*

$$\mathbb{E}_{\boldsymbol{\xi}_i \sim \mathcal{D}_i}[\nabla F(\boldsymbol{\theta}; \boldsymbol{\xi}_i)] = \nabla f_i(\boldsymbol{\theta})$$
$$\mathbb{E}_{\boldsymbol{\xi}_i \sim \mathcal{D}_i}\|\nabla F(\boldsymbol{\theta}; \boldsymbol{\xi}_i) - \nabla f_i(\boldsymbol{\theta})\|^2 \leq \sigma^2, \forall i.$$

Note that our approach eliminates the requirement of gradient dissimilarity bound, typically given by $\frac{1}{N}\sum_i \|\nabla f_i(\boldsymbol{\theta})\|^2 \leq B\|\nabla f(\boldsymbol{\theta})\|^2 + \sigma_h^2, \forall \boldsymbol{\theta}$.

**Theorem 1.** *Let $\{\boldsymbol{\theta}^t\}_{t=1}^T$ be the global iterates generated by ParFreFL. Set $\beta = \frac{\sqrt{SK}}{\sqrt{T}}$, $\eta = \frac{1}{K(SKT)^{1/4}}$, and $\gamma = \frac{(SK)^{1/4}}{T^{3/4}}$. Then, under Assumptions 1 and 2, we have:*

$$\frac{1}{T}\sum_{t=0}^{T-1}\mathbb{E}\left\|\nabla f(\boldsymbol{\theta}^t)\right\|$$
$$\leq \mathcal{O}\left(\frac{\Delta + L}{(SKT)^{1/4}} + \frac{\sigma}{\sqrt{T}} + \frac{S^{1/4}\sigma}{\sqrt{N}(KT)^{1/4}}\right)$$

*where $\Delta := f(\boldsymbol{\theta}^0) - \min_{\boldsymbol{\theta}} f(\boldsymbol{\theta})$.*

**Remark 1.** *According to Theorem 1, ParFreFL preserves the parameter-free properties of PAdaMFed, with all learning rates explicitly determined by system constants $S$, $K$,*

*and $T$, thus eliminating the tedious process of hyperparameter tuning. When $T \geq SK$, i.e., $\frac{\sigma}{\sqrt{T}} \leq \frac{\sigma}{(SKT)^{1/4}}$, the convergence rate is upper bounded by $\mathcal{O}\left(\frac{\Delta + L + \sigma}{(SKT)^{1/4}}\right)$, demonstrating linear speedup with respect to both the number of participating clients and local update rounds.*

**Remark 2.** *Beyond halving communication costs, ParFreFL achieves a tighter bound compared to PAdaMFed, reducing the $\frac{\sqrt{SK}\sigma}{\sqrt{T}}$ term in PAdaMFed to $\frac{\sigma}{\sqrt{T}}$. With this improvement, setting $SK = \mathcal{O}(T)$ enables ParFreFL to achieve an accuracy of $\mathbb{E}\|\nabla f(\boldsymbol{\theta})\| \leq \epsilon$ in $\mathcal{O}(\epsilon^{-2})$ communication rounds, whereas PAdaMFed requires $\mathcal{O}(\epsilon^{-3})$ rounds.*

**Proof Sketch.** Our theoretical analysis begins with the $L$-smoothness property of the loss function $f(\cdot)$, which establishes the following relationship:

$$\frac{1}{T}\sum_t \mathbb{E}\left\|\nabla f\left(\boldsymbol{\theta}^t\right)\right\| \leq \frac{\Delta}{\gamma T} + \frac{2}{T}\sum_t \mathbb{E}\left\|\nabla f\left(\boldsymbol{\theta}^t\right) - \boldsymbol{m}^t\right\|$$
$$+ \frac{2}{T}\sum_t \mathbb{E}\left\|\boldsymbol{m}^t - \boldsymbol{g}^t\right\| + \frac{\gamma L}{2}.$$

Here, the term $\frac{1}{T}\sum_t \mathbb{E}\left\|\nabla f\left(\boldsymbol{\theta}^t\right) - \boldsymbol{m}^t\right\|$ quantifies the discrepancy between the true gradient and the descent direction under full client participation. Meanwhile, the term $\frac{1}{T}\sum_t \mathbb{E}\left\|\boldsymbol{m}^t - \boldsymbol{g}^t\right\|$ captures the error introduced by partial client participation. By carefully controlling these errors, we obtain: $\frac{1}{T}\sum_t \mathbb{E}\left\|\nabla f\left(\boldsymbol{\theta}^t\right) - \boldsymbol{m}^t\right\| \leq \mathcal{O}\left(\frac{L}{(SKT)^{1/4}} + \frac{S^{1/4}\sigma}{\sqrt{N}(KT)^{1/4}}\right)$ and $\frac{1}{T}\sum_t \mathbb{E}\left\|\boldsymbol{m}^t - \boldsymbol{g}^t\right\| \leq \mathcal{O}\left(\frac{\sigma}{\sqrt{T}}\right)$. Combining these results yields the final convergence rate.

## 3. Parameter-Free Compressed FL

In this section, we develop **ComParFreFL**, a compressed variant to further enhance the communication efficiency of parameter-free FL. The procedures of ComParFreFL are presented in Algorithm 2. This algorithm efficiently handles biased compression while maintaining ParFreFL's minimal communication cost through a novel compression mechanism.

A seemingly natural approach is to directly compress the client momentum, the control variate, or their separate increments. However, such designs either require transmitting multiple model-sized quantities or introduce coupled compression errors that are difficult to control under parameter-free normalized updates. Our construction in ComParFreFL instead compresses only the single difference

$$\boldsymbol{\delta}_i^{t+1} = \boldsymbol{m}_i^{t+1} - \boldsymbol{c}_i^t = \boldsymbol{m}_i^{t+1} - \boldsymbol{c}_i^{t-1} - \mathcal{C}(\boldsymbol{\delta}_i^t),$$

---

**Algorithm 2** ComParFreFL: Parameter-Free Compressed Federated Learning

---

1: **Require:** Initial model $\boldsymbol{\theta}^0$, $\boldsymbol{m}_i^{-1} = \frac{1}{K}\sum_{k=0}^{K-1}\nabla F\left(\boldsymbol{\theta}^0; \boldsymbol{\xi}_i^{-1,k}\right)$ and $\boldsymbol{c}_i^{-1} = \boldsymbol{m}_i^{-1}$ for any $i$, $\boldsymbol{c}^{-1} = \frac{1}{N}\sum_i \boldsymbol{c}_i^{-1}$, learning rates $\eta$ and $\gamma$, and momentum parameter $\beta$
2: **for** $t = 0, \cdots, T-1$ **do**
3:   **Central Server:** Uniformly sample clients $\mathcal{S}_t \subseteq \{1, \cdots, N\}$ with $|\mathcal{S}_t| = S$
4:   **for** each client $i \in \mathcal{S}_t$ in parallel **do**
5:     Receive $\boldsymbol{\theta}^t$ and initialize $\boldsymbol{\theta}_i^{t,0} = \boldsymbol{\theta}^t$
6:     **for** $k = 1, \cdots, K$ **do**
7:       Compute $\boldsymbol{m}_i^{t,k} = (1 - \beta)\boldsymbol{m}_i^{t-1} + \beta\nabla F\left(\boldsymbol{\theta}_i^{t,k}; \boldsymbol{\xi}_i^{t,k}\right)$
8:       Update local model $\boldsymbol{\theta}_i^{t,k+1} = \boldsymbol{\theta}_i^{t,k} - \eta\frac{\boldsymbol{m}_i^{t,k}}{\|\boldsymbol{m}_i^{t,k}\|}$
9:     **end for**
10:     Compute $\boldsymbol{m}_i^t = \frac{1}{K}\sum_k \boldsymbol{m}_i^{t,k}$
11:     Compute $\boldsymbol{\delta}_i^t = \boldsymbol{m}_i^t - \boldsymbol{c}_i^{t-1}$
12:     Upload $\widetilde{\boldsymbol{\delta}}_i^t = \mathcal{C}(\boldsymbol{\delta}_i^t)$ to central server
13:     Update $\boldsymbol{c}_i^t = \boldsymbol{c}_i^{t-1} + \widetilde{\boldsymbol{\delta}}_i^t$   ($\boldsymbol{m}_i^t = \boldsymbol{m}_i^{t-1}$ and $\boldsymbol{c}_i^t = \boldsymbol{c}_i^{t-1}$ for $i \notin \mathcal{S}_t$)
14:   **end for**
    **Central server:**
15:   Aggregate control variate $\boldsymbol{c}^t = \boldsymbol{c}^{t-1} + \frac{1}{N}\sum_{i\in\mathcal{S}_t}\widetilde{\boldsymbol{\delta}}_i^t$
16:   Compute $\widetilde{\boldsymbol{g}}^t = \frac{1}{S}\sum_{i\in\mathcal{S}_t}\widetilde{\boldsymbol{\delta}}_i^t + \boldsymbol{c}^{t-1}$
17:   Update global model $\boldsymbol{\theta}^{t+1} = \boldsymbol{\theta}^t - \gamma\frac{\widetilde{\boldsymbol{g}}}{\|\widetilde{\boldsymbol{g}}\|}$
18:   Download $\boldsymbol{\theta}^{t+1}$ to all clients
19: **end for**

---

where $\mathcal{C}(\cdot)$ represents a compression operator, and $\boldsymbol{c}_i^t$ is the local control variate. This update can be equivalently decomposed into two components:

$$\boldsymbol{\delta}_i^{t+1} = \underbrace{\boldsymbol{m}_i^{t+1} - \boldsymbol{m}_i^t}_{\textbf{Momentum increment}} + \underbrace{\boldsymbol{\delta}_i^t - \mathcal{C}(\boldsymbol{\delta}_i^t)}_{\textbf{Error feedback}}.$$

This form is crucial because it merges the momentum increment and the error-feedback residual into one transmitted quantity. As a result, ComParFreFL preserves the one-vector communication cost of ParFreFL, keeps the compressed control-variate recursion closed, and allows the effect of biased compression to be confined to lower-order terms in the convergence bound without introducing compression-ratio-dependent stepsizes.

The momentum increment is given by:

$$\boldsymbol{m}_i^t - \boldsymbol{m}_i^{t-1} = \beta\left(\frac{1}{K}\sum_k \nabla F\left(\boldsymbol{\theta}_i^{t,k}; \boldsymbol{\xi}_i^{t,k}\right) - \boldsymbol{m}_i^{t-1}\right),$$

where $\beta$ is the momentum coefficient. The recursive na-

*Table 1.* Comparison of compressed FL algorithms with biased compression. **D. H.** denotes allowing arbitrary data heterogeneity; $\epsilon$ is a target for the stationarity $\mathbb{E}\|f(\boldsymbol{\theta})\| \leq \epsilon$; **Com. R.** is the number of communication rounds required to achieve a desired accuracy $\epsilon$; **Trans. Size Up./down.** The parameter size of uplink/downlink transmission; **Step. Para.**: problem-specific parameters related to the stepsize restrictions; $d$: model size; $N, S, K, T$: the number of total clients, participating clients, local update rounds, and communication rounds, respectively.

| Algorithm | D. H. | Com. R.[1] | Trans. Size Up./down. | Stepsize Restrictions | Step. Para. |
|---|---|---|---|---|---|
| **LOCAL-SGD-C** (Gao et al., 2021) | ✗ | $\frac{K}{(1-q)^2\epsilon^2}$ | $d/d$ | $\eta \leq \frac{\sqrt{1+2(a_1+a_2+16)K^2}-1}{(a_1+a_2+16)K^2 L}$ $a_1 = \frac{384(1+q^2)q^2}{(1-q^2)^4},\ a_2 = \frac{48q^2}{(1-q^2)^2}$ | $q, L$ |
| **FED-EF-SGD** (Li & Li, 2023) | ✗ | $\frac{1}{(1-q)^2\epsilon^2}$ | $d/d$ | $\eta \leq \frac{1}{2KL\max\{4,\eta(C+1)\}}$ $C = 2 + \frac{4q^2}{(1-q^2)^2}$ | $q, L$ |
| **FED-EF-ASM** (Li & Li, 2023) | ✗ | $\frac{1}{(1-q)^2\epsilon^2}$ | $d/d$ | $\eta \leq \frac{\sqrt{\delta}}{8KL}\min\left\{\frac{1}{\sqrt{\delta}}, \frac{2(1-q^2)L}{(1+q^2)^{3/2}G},\right.$ $\left.\frac{1}{\max\{16,32C^2\}\gamma}, \frac{1}{3\gamma^{1/3}}\right\}^2$ $C = \frac{\beta_1}{1-\beta_1} + \frac{2q}{1-q^2}$ | $q, L, G$ |
| **SCAFCOM** Huang et al. (2024a) | ✓ | $\frac{1}{(1-q)\epsilon^2}$ | $d/2d$ | $\eta K L \leq \sqrt{\frac{\beta(1-q)^2}{36e^2 N\left(189(1-q)^2+306\beta^2\right)}}$ $\gamma\eta KL = \left(\frac{20N}{\beta S} + \frac{28N}{(1-q)S}\right)^{-1}$ $\beta = \left(1 + \left(\frac{TS\sigma^2}{N^2 KL\Delta}\right)^{1/2} +\right.$ $\left.\left(\frac{TS\sigma^2}{NK(1-q)L\Delta}\right)^{1/3}+\left(\frac{TS\sigma^2}{NK(1-q)^2 L\Delta}\right)^{1/4}\right)^{-1}$ | $q, L, \sigma^2, \Delta$ |
| **COMPARFREFL** (This Paper) | ✓ | $\frac{q^2 S}{(1-q)^2\epsilon^2}$[3] | $d/d$ | $\beta = \frac{\sqrt{SK}}{\sqrt{T}},\ \eta = \frac{1}{K(SKT)^{1/4}},\ \gamma = \frac{(SK)^{1/4}}{T^{3/4}}$ | None |

[1] We convert the bound in terms of $\frac{1}{T}\sum_{t=1}^{T-1}\mathbb{E}\|\nabla f(\boldsymbol{\theta})\|^2$ to $\frac{1}{T}\sum_{t=1}^{T-1}\mathbb{E}\|\nabla f(\boldsymbol{\theta})\|$, based on Jensen's inequality that $\frac{1}{T}\sum_{t=1}^{T-1}\mathbb{E}\|\nabla f(\boldsymbol{\theta})\| \leq \sqrt{\frac{1}{T}\sum_{t=1}^{T-1}\mathbb{E}\|\nabla f(\boldsymbol{\theta})\|^2}$

[2] Here, $\delta$ and $\beta_1$ are predefined hyperparameters and $G$ is a constant defined as $\|\nabla F(\boldsymbol{\theta};\boldsymbol{\xi})\| \leq G, \forall\boldsymbol{\theta},\boldsymbol{\xi}$

[3] Calculated when all clients share a common compression rate $q$, then $\alpha = \frac{q}{1-q}$

ture of momentum ensures this term diminishes as the algorithm approaches convergence. By appropriately selecting the coefficient $\beta$, the momentum increment can be effectively controlled, further reducing compression error.

### 3.1. Theoretical Results

**Assumption 3** ($q^2$-Contractive Compressor). *For any client $i \in [N]$, there exists $q_i \in [0,1)$ such that for any input $\boldsymbol{\theta} \in \mathbb{R}^d$, its compressor $\mathcal{C}_i : \mathbb{R}^d \to \mathbb{R}^d$ satisfies:*

$$\mathbb{E}\|\mathcal{C}_i(\boldsymbol{\theta}) - \boldsymbol{\theta}\|^2 \leq q_i^2\|\boldsymbol{\theta}\|^2$$

*where the expectation is taken over the randomness of the compressor $\mathcal{C}_i$.*

**Theorem 2.** *Let $\{\boldsymbol{\theta}^t\}_{t=1}^T$ be the global iterates generated by ComParFreFL. Set $\beta = \frac{\sqrt{SK}}{\sqrt{T}}$, $\eta = \frac{1}{K(SKT)^{1/4}}$, and $\gamma = \frac{(SK)^{1/4}}{T^{3/4}}$. Then, under Assumptions 1-3, we have:*

$$\frac{1}{T}\sum_{t=0}^{T-1}\mathbb{E}\left\|\nabla f\left(\boldsymbol{\theta}^t\right)\right\| \leq \mathcal{O}\left(\frac{\Delta+L}{(SKT)^{1/4}} + \frac{\alpha\sqrt{S}\sigma}{\sqrt{T}} + \frac{\sigma}{\sqrt{T}}\right)$$

*where $\alpha := \sqrt{\frac{1}{N}\sum_i \frac{q_i^4}{(1-q_i)^2}} + \sqrt{\frac{1}{N}\sum_i q_i^2}$.*

**Remark 3.** *Theorem 2 establishes that compressed transmission affects only the subdominant terms $\mathcal{O}\left(\frac{\alpha\sqrt{S}\sigma}{\sqrt{T}}\right)$ while preserving the leading term in the convergence bound of ComParFreFL. Here, $\alpha$ denotes the average compression rate, where larger values indicate higher compression and necessitate additional iterations for convergence. In contrast, this compression factor often inevitably degrades all terms in the convergence bound of existing state-of-the-art approaches, including LOCAL-SGD-C (Gao et al., 2021), FED-EF-SGD, and FED-EF-ASM (Li & Li, 2023).*

**Remark 4.** *The stepsize setting of ComParFreFL is also independent of the compression ratio, which, to the best of our knowledge, is the first instance of such robustness in the compressed FL literature. This robustness enables clients to dynamically adjust their communication schemes in response to changing network conditions without requiring any system reconfiguration, thereby enhancing the practicality and adaptability of our approach.*

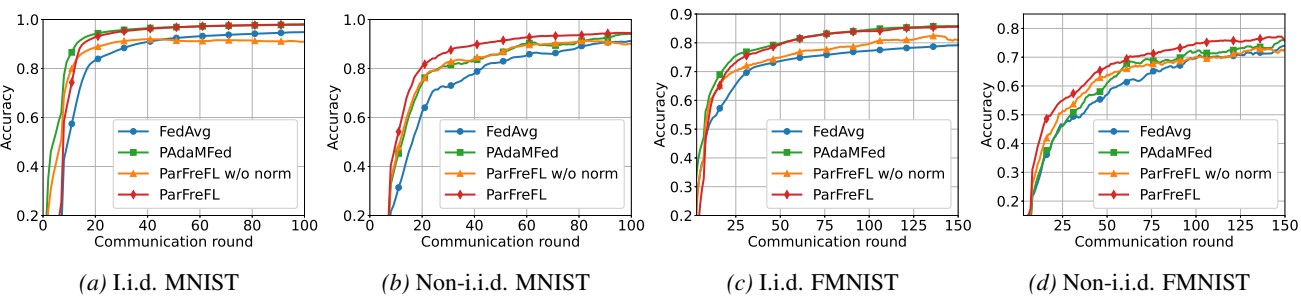

*Figure 1.* Test accuracy versus communication round of ParFreFL (Algorithm 1).

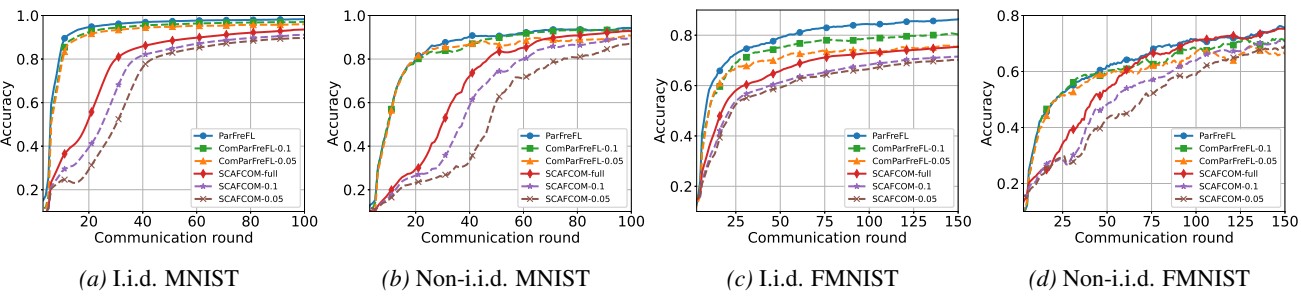

*Figure 2.* Test accuracy versus communication round of ComParFreFL (Algorithm 2).

### 3.2. Comparisons with Existing Compressed FL Approaches

In Table 1, we compare ComParFreFL with representative compressed FL approaches applicable to biased compressors. A comprehensive literature review appears in the Appendix A due to space limitations. As shown in Table 1, earlier approaches (LOCAL-SGD-C (Gao et al., 2021), FED-EF-SGD, and FED-EF-ASM (Li & Li, 2023)) struggle with heterogeneous data distributions. While SCAFCOM Huang et al. (2024a) overcomes this limitation through a carefully designed momentum mechanism—a strategy we also leverage—it requires transmitting two model-sized parameters in downlink communication. Our method enhances communication efficiency by requiring only one model-sized parameter for both uplink and downlink transmission.

SCAFCOM demonstrates superior communication complexity among the compared methods. Our approach introduces a slowing coefficient $q^2 S$ in the communication complexity. Since $q < 1$ is typically small in practical compression schemes (e.g., 1-bit SGD, Top-$K$ sparsification) and the number of participating clients $S$ remains controllable, this coefficient presents a manageable trade-off in practice.

The most significant contribution of our approach is its parameter-free property. As evidenced in the final two columns of Table 1, existing methods require complex stepsize configurations to ensure convergence, with SCAFCOM particularly dependent on multiple problem-specific

parameters $(q, L, \sigma^2, \Delta)$. These dependencies substantially complicate deployment and reduce robustness in dynamic environments. In contrast, our ComParFreFL method operates entirely independent of problem-specific parameters—including the compression ratio—making it the first such algorithm to our knowledge. This parameter-free design significantly enhances accessibility and deployability in dynamic, resource-constrained federated environments.

## 4. Numerical Experiments

This section presents empirical validation of our proposed methods. The experimental results focus on two primary objectives: (1) demonstrating the problem-parameter-free property of ParFreFL through comparisons with PAdaMFed (Yan et al., 2024) and FedAvg (McMahan et al., 2017); and (2) evaluating the communication efficiency of ComParFreFL against the state-of-the-art SCAFCOM algorithm (Huang et al., 2024a). For all figures, the learning rates of our approaches, ParFreFL and ComParFreFL, are determined based on theoretical guidance (given in Theorem 1 and Theorem 2, respectively) while the learning rates of all baselines are optimized via grid search. Additional experimental details and simulation results are provided in Appendix D.

Figure 1 presents test accuracy versus communication rounds for ParFreFL under full precision on MNIST and FMNIST datasets with both i.i.d. and non-i.i.d. data. Despite requiring only half the per-round communication cost, ParFreFL achieves comparable accuracy to PAdaMFed on

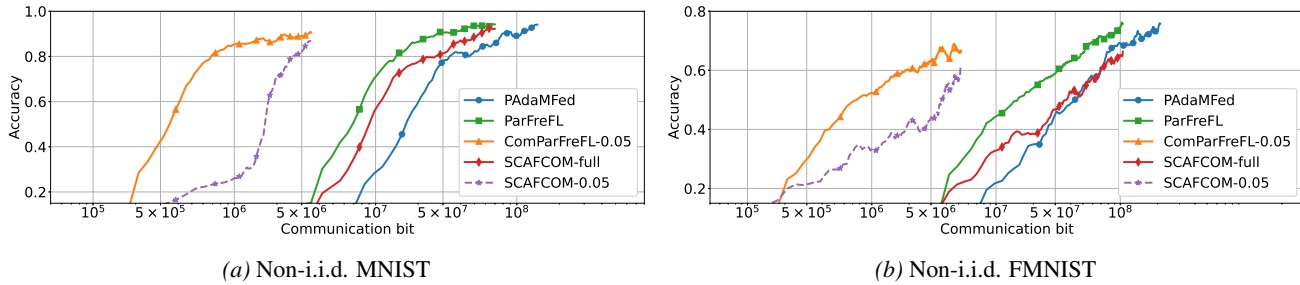

*(a)* Non-i.i.d. MNIST        *(b)* Non-i.i.d. FMNIST

*Figure 3.* Test accuracy versus communication bits.

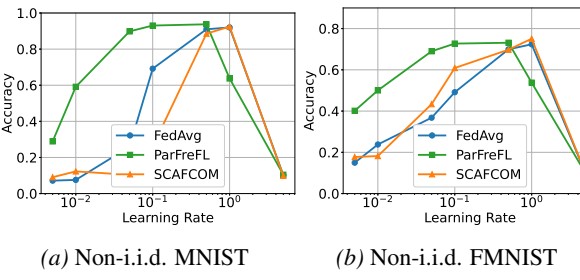

*(a)* Non-i.i.d. MNIST     *(b)* Non-i.i.d. FMNIST

*Figure 4.* Test accuracy versus learning rate of ParFreFL.

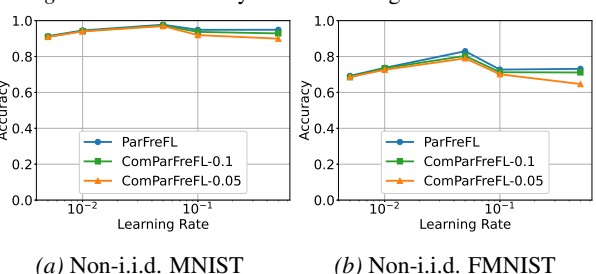

*(a)* Non-i.i.d. MNIST     *(b)* Non-i.i.d. FMNIST

*Figure 5.* Test accuracy versus learning rate under different compression ratios.

i.i.d. data and superior performance in non-i.i.d. settings. Both parameter-free methods consistently outperform the optimally-tuned FedAvg baseline, demonstrating the effectiveness of parameter-free approaches in FL. These results confirm that our method preserves convergence guarantees while substantially reducing communication overhead, establishing its viability for bandwidth-constrained FL applications.

Figure 2 illustrates the test accuracy versus communication rounds for ComParFreFL with different compression ratios. We adopt Top-$k$ sparsification where ComParFreFL-0.1 denotes transmission of only the top $10\%$ entries by absolute value. On the MNIST dataset, ComParFreFL closely tracks the uncompressed baseline despite data heterogeneity at both 0.1 and 0.05 compression ratios. For FMNIST dataset, marginal performance degradation is observed under aggressive compression. Across all compression levels, ComParFreFL maintains competitive convergence rates compared to SCAFCOM, demonstrating robust

performance under significant communication constraints.

Figure 3 presents test accuracy versus total communication bits for both compressed and full-precision algorithms. We evaluate SCAFCOM, ParFreFL, and their compressed variants with Top-0.05 sparsification, alongside full-precision PAdaMFed. The total communication cost only accounts for uplink transmission, yielding $d$ parameters per round for ParFreFL, $d$ for SCAFCOM, and $2d$ for PAdaMFed, where $d$ denotes the model dimension. ComParFreFL-0.05 achieves competitive accuracy while requiring an order of magnitude fewer communication bits than full-precision methods. Although SCAFCOM-0.05 similarly reduces communication overhead, it demonstrates slower convergence relative to ComParFreFL under equivalent bit constraints. These results validate the communication efficiency of our compressed parameter-free approach, demonstrating an effective balance between model performance and bandwidth limitations in FL.

Figure 4 and Figure 5 examine learning rate sensitivity across different algorithms and compression levels. Figure 4 demonstrates that ParFreFL maintains consistently high accuracy across a wide range of learning rates (0.05 to 0.5), significantly outperforming the narrow optimal ranges of FedAvg and SCAFCOM, which suffer severe performance degradation outside their tuned values. Figure 5 reveals that this robustness is preserved under compression: ComParFreFL with both 0.1 and 0.05 compression ratios exhibits similar learning rate insensitivity to the full-precision version, confirming that our compressed variant retains the key advantage of parameter-free optimization while reducing communication costs.

## 5. Conclusions

This paper addresses the two most critical barriers to FL adoption: excessive communication costs and the need for expert hyperparameter optimization. By simultaneously reducing communication overhead and removing problem-specific hyperparameter tuning requirements, including compression ratios, our methods make FL more practical for resource-constrained environments.

## Impact Statement

This paper presents work whose goal is to advance the field of Machine Learning. There are many potential societal consequences of our work, none of which we feel must be specifically highlighted here.

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

# A. Related Work

**Adaptive Gradient Methods.** The challenge of manual learning rate tuning motivated the development of adaptive optimizers. This line of work began with AdaGrad (Duchi et al., 2011), which introduced per-parameter learning rates scaled by accumulated squared gradients. To mitigate AdaGrad's aggressive learning rate decay, RMSProp (Hinton et al., 2012) and Adam (Kingma & Ba, 2014) adopted exponential moving averages, with Adam also incorporating momentum to become the de facto standard for deep learning. Numerous variants have since been proposed to enhance convergence guarantees (AMSGrad (Reddi et al., 2019)), improve stability in sparse settings (Yogi (Zaheer et al., 2018)), reduce memory footprints for large models (AdaFactor (Shazeer & Stern, 2018)), or discover novel update rules via program search (Lion (Chen et al., 2023)). Despite their success, these methods remain sensitive to hyperparameter choices, which can significantly impact both convergence speed and final model generalization (Wilson et al., 2017).

**Parameter-Free Optimization.** Parameter-free algorithms emerged from online learning theory to achieve optimal regret bounds without requiring prior knowledge of problem constants. Coin-betting algorithms (Orabona & Pál, 2016) and DoG (Cutkosky & Orabona, 2018) adapt step sizes using gradient history, with recent extensions like DoWG (Khaled et al., 2023) adding preconditioning to enhance convergence. However, these methods face significant communication bottlenecks in distributed settings where their dependence on global statistics incurs $\mathcal{O}(N^2)$ communication overhead for $N$ workers (Cutkosky & Orabona, 2018; Khaled et al., 2023) . In parallel, normalization techniques control update magnitudes to ensure stability. This includes gradient clipping to prevent exploding gradients (Pascanu et al., 2013) and layer-wise adaptive scaling in LARS (You et al., 2017) and LAMB (You et al., 2019), which have been critical for large-batch distributed training. The empirical success of these methods is supported by theory proving dimension-free convergence for normalized gradient descent (Yang et al., 2023; Cutkosky & Mehta, 2020) and the surprising effectiveness of extreme approaches like SignSGD (Bernstein et al., 2018).

**Adaptive Optimization in Federated Learning.** Translating adaptive methods to federated settings presents unique challenges due to data heterogeneity and communication constraints. FedAdam and FedYogi (Reddi et al., 2020) directly apply server-side adaptive optimization but require transmitting additional optimizer states. FedOpt (Asad et al., 2020) provides a general framework for incorporating various adaptive server optimizers. Local adaptive methods like FedLion (Tang & Chang, 2024) maintain client-side adaptive states to reduce communication while preserving benefits of adaptivity. Recent work explores hybrid approaches: FedAGM (Ba et al., 2024) combines global momentum with local adaptivity, while FAdamGC (Chen et al., 2025) uses adaptive gradient scaling to handle non-IID data. While these methods demonstrate that appropriately designed adaptation mechanisms can substantially improve convergence in heterogeneous federated environments, they remain dependent on problem-specific hyperparameter tuning that significantly influences both convergence and generalization performance.

The recent breakthrough of PAdaMFed (Yan et al., 2024) has pioneered the investigation of parameter-free methods for federated learning, eliminating the requirement for manual tuning of problem-specific hyperparameters. Subsequent work (Yan et al., 2025) has further demonstrated its effectiveness in handling arbitrarily heterogeneous data. However, both approaches incur twice the communication overhead compared to conventional FedAvg-based algorithms (McMahan et al., 2017).

**Compressed Federated Learning.** The prohibitive communication overhead in FL (Zhong et al., 2026) has motivated extensive research into compression-aware optimization algorithms. Early work by Reisizadeh et al. (2020) demonstrated that periodic model averaging combined with aggressive quantization could reduce communication costs while preserving convergence guarantees. Building upon this foundation, Haddadpour et al. (2021) established theoretical frameworks demonstrating that properly designed compression schemes can achieve linear speedup in federated settings without compromising accuracy. Subsequently, Li & Li (2023) advanced error feedback techniques to address the dual challenges of non-IID data distributions and irregular client availability—key practical constraints in real-world deployments.

Recent theoretical advances have further refined our understanding of compression in federated optimization. Fatkhullin et al. (2023) demonstrated that strategically incorporating momentum into communication compression can effectively mitigate the effects of biased compression. Building on these insights, Huang et al. (2024a) proposed SCAFCOM, which simultaneously handles unbiased compression while maintaining robustness to data heterogeneity and partial client participation. However, these existing methods require careful tuning of learning rates to ensure convergence, limiting their practical applicability. In contrast, our proposed ComParFreFL algorithm retains the advantages of SCAFCOM while operating independently of problem-specific parameters, including the compression ratio, thereby enhancing the applicability of FL in dynamic, resource-constrained environments.

# B. Proof of Theorem 1

Throughout the analysis, we use the following notation for summations:

- $\sum_i$ denotes summation over all clients $i \in \{1, \dots, N\}$;

- $\sum_{i \in \mathcal{S}_t}$ denotes summation over selected clients in multiset $\mathcal{S}_t$;

- $\sum_k$ denotes summation over local steps $k \in \{0, \dots, K-1\}$;

- $\sum_t$ denotes summation over global rounds $t \in \{0, \dots, T-1\}$.

Based on the $L$-smoothness of $f(\cdot)$ in Assumption 1 and the update rule of $\boldsymbol{\theta}^t$, we have

$$
\begin{aligned}
f\left(\boldsymbol{\theta}^{t+1}\right) - f\left(\boldsymbol{\theta}^t\right) &\leq \nabla f\left(\boldsymbol{\theta}^t\right)^\top \left(\boldsymbol{\theta}^{t+1} - \boldsymbol{\theta}^t\right) + \frac{L}{2}\left\|\boldsymbol{\theta}^{t+1} - \boldsymbol{\theta}^t\right\|^2 \\
&\leq -\gamma \nabla f\left(\boldsymbol{\theta}^t\right)^\top \frac{\boldsymbol{g}^t}{\|\boldsymbol{g}^t\|} + \frac{\gamma^2 L}{2} \\
&= -\gamma \left(\nabla f\left(\boldsymbol{\theta}^t\right) - \boldsymbol{g}^t\right)^\top \frac{\boldsymbol{g}^t}{\|\boldsymbol{g}^t\|} - \gamma \left(\boldsymbol{g}^t\right)^\top \frac{\boldsymbol{g}^t}{\|\boldsymbol{g}^t\|} + \frac{\gamma^2 L}{2} \\
&\leq \gamma \left\|\nabla f\left(\boldsymbol{\theta}^t\right) - \boldsymbol{g}^t\right\| - \gamma \left\|\boldsymbol{g}^t\right\| + \frac{\gamma^2 L}{2} \\
&\leq 2\gamma \left\|\nabla f\left(\boldsymbol{\theta}^t\right) - \boldsymbol{g}^t\right\| - \gamma \left\|\nabla f\left(\boldsymbol{\theta}^t\right)\right\| + \frac{\gamma^2 L}{2},
\end{aligned}
\tag{1}
$$

where the last inequality is based on $\gamma \left\|\nabla f(\boldsymbol{\theta}^t)\right\| - \gamma \|\boldsymbol{g}^t\| \leq \gamma \left\|\nabla f(\boldsymbol{\theta}^t) - \boldsymbol{g}^t\right\|$.

Define $\boldsymbol{h}_i^t := \frac{1}{K} \sum_k \nabla F\left(\boldsymbol{\theta}_i^{t,k}; \boldsymbol{\xi}_i^{t,k}\right)$, $\boldsymbol{h}^t := \frac{1}{N} \sum_i \boldsymbol{h}_i^t$, and

$$
\boldsymbol{m}^t := \frac{1}{N} \sum_i \boldsymbol{m}_i^t = \frac{1}{N} \sum_i \left((1-\beta)\boldsymbol{m}_i^{t-1} + \beta \boldsymbol{h}_i^t\right) = (1-\beta)\boldsymbol{m}^{t-1} + \beta \boldsymbol{h}^t.
$$

In ParFreFL, the global descent direction $\boldsymbol{g}^t$ is updated as:

$$
\boldsymbol{g}^t = \frac{1}{S} \sum_{i \in \mathcal{S}_t} \left(\boldsymbol{c}_i^t - \boldsymbol{c}_i^{t-1}\right) + \boldsymbol{c}^{t-1} = \frac{1}{S} \sum_{i \in \mathcal{S}_t} \left((1-\beta)\boldsymbol{m}_i^{t-1} + \beta \boldsymbol{h}_i^t - \boldsymbol{c}_i^{t-1}\right) + \boldsymbol{c}^{t-1},
$$

Thus, $\boldsymbol{m}^t$ can be considered the global descent vector under full client participation since $\frac{1}{N} \sum_i \boldsymbol{c}_i^t = \boldsymbol{c}^t$ always holds.

Using the triangle inequality, we can bound the distance between the true gradient and the global descent direction as follows:

$$
\mathbb{E}\left\|\nabla f\left(\boldsymbol{\theta}^t\right) - \boldsymbol{g}^t\right\| \leq \mathbb{E}\left\|\nabla f\left(\boldsymbol{\theta}^t\right) - \boldsymbol{m}^t\right\| + \mathbb{E}\left\|\boldsymbol{m}^t - \boldsymbol{g}^t\right\|,
$$

where the first term quantifies the discrepancy between the true gradient and the descent direction under full participation, and the second term captures the error introduced by partial client participation.

Plugging the above result into Equation (1) and summing up over $t$ gives

$$
\frac{1}{T} \sum_t \mathbb{E}\left\|\nabla f\left(\boldsymbol{\theta}^t\right)\right\| \leq \frac{\Delta}{\gamma T} + \frac{2}{T} \sum_t \mathbb{E}\left\|\nabla f\left(\boldsymbol{\theta}^t\right) - \boldsymbol{m}^t\right\| + \frac{2}{T} \sum_t \mathbb{E}\left\|\boldsymbol{m}^t - \boldsymbol{g}^t\right\| + \frac{\gamma L}{2},
$$

where $\Delta := f(\boldsymbol{\theta}^0) - \min_{\boldsymbol{\theta}} f(\boldsymbol{\theta})$.

From Lemma 2, we know that

$$
\frac{1}{T} \sum_t \mathbb{E}\left\|\nabla f\left(\boldsymbol{\theta}^t\right) - \boldsymbol{m}^t\right\| \leq \frac{2}{\beta T}\left(\frac{\sigma}{\sqrt{NK}} + \beta \eta K L\right) + \frac{2\gamma L}{\beta} + \eta K L + \frac{\sqrt{\beta}\sigma}{\sqrt{NK}}.
$$

By Lemma 3, we have

$$\frac{1}{T}\sum_t \mathbb{E}\left\|\boldsymbol{m}^t - \boldsymbol{g}^t\right\| \leq \frac{2\beta\sigma}{\sqrt{SK}} + \frac{2}{\sqrt{S}}\beta\eta KL + \frac{3}{\sqrt{S}}\gamma L + \frac{2\sqrt{3}}{T\sqrt{S}}\left(\frac{\sigma}{\sqrt{SK}} + \beta\eta KL\right).$$

Thus, we have

$$\frac{1}{T}\sum_t \mathbb{E}\left\|\nabla f\left(\boldsymbol{\theta}^t\right)\right\| \leq \frac{\Delta}{\gamma T} + 2\left(\frac{2}{T} + 1 + \frac{2\beta}{\sqrt{S}} + \frac{2\sqrt{3}\beta}{T\sqrt{S}}\right)\eta KL + \left(\frac{4}{\beta} + \frac{6}{\sqrt{S}} + \frac{1}{2}\right)\gamma L$$
$$+ 2\left(\frac{2}{\beta T} + \sqrt{\beta} + 2\beta\frac{\sqrt{N}}{\sqrt{S}} + \frac{2\sqrt{3}\sqrt{N}}{ST}\right)\frac{\sigma}{\sqrt{NK}}.$$

Let $\beta = \frac{\sqrt{SK}}{\sqrt{T}}$, $\eta = \frac{1}{K(SKT)^{1/4}}$, and $\gamma = \frac{(SK)^{1/4}}{T^{3/4}}$, then

$$\frac{1}{T}\sum_t \mathbb{E}\left\|\nabla f\left(\boldsymbol{\theta}^t\right)\right\| \leq \mathcal{O}\left(\frac{\Delta + L}{(SKT)^{1/4}} + \frac{S^{1/4}\sigma}{\sqrt{N}(KT)^{1/4}} + \frac{\sigma}{\sqrt{T}}\right).$$

### B.1. Proof of Technical Lemmas

**Lemma 1.** *Given vectors* $\boldsymbol{\omega}_1, \cdots, \boldsymbol{\omega}_N \in \mathbb{R}^d$ *and* $\overline{\boldsymbol{\omega}} = \frac{1}{N}\sum_{i=1}^N \boldsymbol{\omega}_i$, *if we sample* $\mathcal{S} \subset \{1, \cdots, N\}$ *uniformly randomly such that* $|\mathcal{S}| = S$, *then it holds that*

$$\mathbb{E}\left[\left\|\frac{1}{S}\sum_{i\in\mathcal{S}}\boldsymbol{\omega}_i\right\|^2\right] \leq \|\overline{\boldsymbol{\omega}}\|^2 + \frac{1}{SN}\sum_{i=1}^N \|\boldsymbol{\omega}_i - \overline{\boldsymbol{\omega}}\|^2.$$

*Proof.* Letting $\mathbb{1}\{i \in \mathcal{S}\}$ be the indicator for the event $i \in \mathcal{S}$, we prove this lemma by direct calculation as follows:

$$\mathbb{E}\left[\left\|\frac{1}{S}\sum_{i\in\mathcal{S}}\boldsymbol{\omega}_i\right\|^2\right] = \mathbb{E}\left[\left\|\frac{1}{S}\sum_{i=1}^N \boldsymbol{\omega}_i \mathbb{1}\{i\in\mathcal{S}\}\right\|^2\right]$$
$$= \frac{1}{S^2}\mathbb{E}\left[\left(\sum_i \|\boldsymbol{\omega}_i\|^2 \mathbb{1}\{i\in\mathcal{S}\} + 2\sum_{i<j}\boldsymbol{\omega}_i^\top\boldsymbol{\omega}_j \mathbb{1}\{i,j\in\mathcal{S}\}\right)\right]$$
$$= \frac{1}{SN}\sum_{i=1}^N \|\boldsymbol{\omega}_i\|^2 + \frac{1}{S^2}\frac{S(S-1)}{N(N-1)}2\sum_{i<j}\boldsymbol{\omega}_i^\top\boldsymbol{\omega}_j$$
$$= \frac{1}{SN}\sum_{i=1}^N \|\boldsymbol{\omega}_i\|^2 + \frac{1}{S^2}\frac{S(S-1)}{N(N-1)}\left(\left\|\sum_{i=1}^N \boldsymbol{\omega}_i\right\|^2 - \sum_{i=1}^N \|\boldsymbol{\omega}_i\|^2\right)$$
$$= \frac{N-S}{S(N-1)}\frac{1}{N}\sum_{i=1}^N \|\boldsymbol{\omega}_i\|^2 + \frac{N(S-1)}{S(N-1)}\|\overline{\boldsymbol{\omega}}\|^2$$
$$= \frac{N-S}{S(N-1)}\frac{1}{N}\sum_{i=1}^N \|\boldsymbol{\omega}_i - \overline{\boldsymbol{\omega}}\|^2 + \|\overline{\boldsymbol{\omega}}\|^2$$
$$\leq \frac{1}{SN}\sum_{i=1}^N \|\boldsymbol{\omega}_i - \overline{\boldsymbol{\omega}}\|^2 + \|\overline{\boldsymbol{\omega}}\|^2,$$

where the last inequality uses the fact that $\frac{N-S}{N-1} \leq 1$ for any nonempty set $\mathcal{S}$. $\square$

**Lemma 2.** *For any t, the discrepancies between the true gradients and the actual descent directions in ParFreFL are upper bounded by:*

i) $\mathbb{E}\left\|\nabla f\left(\boldsymbol{\theta}^t\right)-\boldsymbol{m}^t\right\| \leq \left(1-\frac{\beta}{2}\right)^t \left(\frac{\sigma}{\sqrt{NK}}+\beta\eta KL\right)+\left(\frac{2\gamma L}{\beta}+\eta KL+\frac{\sqrt{\beta}\sigma}{\sqrt{NK}}\right)$;

ii) $\mathbb{E}\left\|\nabla f_i\left(\boldsymbol{\theta}^t\right)-\boldsymbol{m}_i^t\right\|^2 \leq (1-\beta)^t \left(\frac{\sigma^2}{K}+\beta^2\eta^2 K^2 L^2\right)+\left(\frac{2\gamma^2 L^2}{\beta^2}+\eta^2 K^2 L^2+\frac{\beta\sigma^2}{K}\right)$ *for any $i$.*

*Proof.* First, we have

$$\mathbb{E}\left\|\nabla f\left(\boldsymbol{\theta}^t\right)-\boldsymbol{m}^t\right\|^2 \tag{2}$$

$$= \mathbb{E}\left\|(1-\beta)\left(\nabla f\left(\boldsymbol{\theta}^t\right)-\boldsymbol{m}^{t-1}\right)-\beta\left(\nabla f\left(\boldsymbol{\theta}^t\right)-\frac{1}{NK}\sum_{i,k}\left(\nabla F\left(\boldsymbol{\theta}_i^{t,k};\boldsymbol{\xi}_i^{t,k}\right)\mp\nabla f_i\left(\boldsymbol{\theta}_i^{t,k}\right)\right)\right)\right\|^2$$

$$= \mathbb{E}\left\|(1-\beta)\left(\nabla f\left(\boldsymbol{\theta}^t\right)\mp\nabla f\left(\boldsymbol{\theta}^{t-1}\right)-\boldsymbol{m}^{t-1}\right)-\beta\left(\nabla f\left(\boldsymbol{\theta}^t\right)-\frac{1}{NK}\sum_{i,k}\nabla f_i\left(\boldsymbol{\theta}_i^{t,k}\right)\right)\right\|^2+\frac{\beta^2\sigma^2}{NK}$$

$$\overset{(a)}{\leq} (1-\beta)\mathbb{E}\left\|\nabla f\left(\boldsymbol{\theta}^{t-1}\right)-\boldsymbol{m}^{t-1}\right\|^2$$

$$+\frac{1}{\beta}\mathbb{E}\left\|(1-\beta)\left(\nabla f\left(\boldsymbol{\theta}^t\right)-\nabla f\left(\boldsymbol{\theta}^{t-1}\right)\right)-\beta\left(\nabla f\left(\boldsymbol{\theta}^t\right)-\frac{1}{NK}\sum_{i,k}\nabla f_i\left(\boldsymbol{\theta}_i^{t,k}\right)\right)\right\|^2+\frac{\beta^2\sigma^2}{NK}$$

$$\leq (1-\beta)\mathbb{E}\left\|\nabla f\left(\boldsymbol{\theta}^{t-1}\right)-\boldsymbol{m}^{t-1}\right\|^2+2\frac{(1-\beta)^2 L^2}{\beta}\mathbb{E}\left\|\boldsymbol{\theta}^t-\boldsymbol{\theta}^{t-1}\right\|^2+2\beta L^2\mathbb{E}\left\|\boldsymbol{\theta}^t-\boldsymbol{\theta}_i^{t,k}\right\|^2+\frac{\beta^2\sigma^2}{NK}$$

$$\leq (1-\beta)\mathbb{E}\left\|\nabla f\left(\boldsymbol{\theta}^{t-1}\right)-\boldsymbol{m}^{t-1}\right\|^2+\frac{2\gamma^2 L^2}{\beta}+\frac{2}{3}\beta\eta^2 K^2 L^2+\frac{\beta^2\sigma^2}{NK}$$

$$\leq (1-\beta)^t\mathbb{E}\left\|\nabla f\left(\boldsymbol{\theta}^0\right)-\boldsymbol{m}^0\right\|^2+\left(\frac{2\gamma^2 L^2}{\beta}+\frac{2}{3}\beta\eta^2 K^2 L^2+\frac{\beta^2\sigma^2}{NK}\right)\sum_{\tau=0}^{t-1}(1-\beta)^\tau$$

$$\leq (1-\beta)^t\left(\frac{\sigma^2}{NK}+\beta^2\eta^2 K^2 L^2\right)+\left(\frac{2\gamma^2 L^2}{\beta^2}+\eta^2 K^2 L^2+\frac{\beta\sigma^2}{NK}\right), \tag{3}$$

where $(a)$ uses the inequality that $\mathbb{E}\|\boldsymbol{a}+\boldsymbol{b}\|^2 \leq \left(1+\frac{\beta}{1-\beta}\right)\mathbb{E}\|\boldsymbol{a}\|^2+\left(1+\frac{1-\beta}{\beta}\right)\mathbb{E}\|\boldsymbol{b}\|^2$ for any vectors $\boldsymbol{a}$ and $\boldsymbol{b}$, and the last inequality is based on the following result:

$$\mathbb{E}\left\|\nabla f\left(\boldsymbol{\theta}^0\right)-\boldsymbol{m}^0\right\|^2 = \mathbb{E}\left\|\nabla f\left(\boldsymbol{\theta}^0\right)-\frac{1}{NK}\sum_{i,k}\left((1-\beta)\nabla F\left(\boldsymbol{\theta}^0;\boldsymbol{\xi}_i^{-1,k}\right)+\beta\nabla F\left(\boldsymbol{\theta}_i^{0,k};\boldsymbol{\xi}_i^{0,k}\right)\right)\right\|^2$$

$$\leq (1-\beta)^2\mathbb{E}\left\|\nabla f\left(\boldsymbol{\theta}^0\right)-\frac{1}{NK}\sum_{i,k}\nabla F\left(\boldsymbol{\theta}^0;\boldsymbol{\xi}_i^{-1,k}\right)\right\|^2$$

$$+\beta^2\mathbb{E}\left\|\frac{1}{N}\sum_i\nabla f_i\left(\boldsymbol{\theta}^0\right)\mp\frac{1}{NK}\sum_{i,k}f_i\left(\boldsymbol{\theta}_i^{0,k}\right)-\frac{1}{NK}\sum_{i,k}\nabla F\left(\boldsymbol{\theta}_i^{0,k};\boldsymbol{\xi}_i^{-1,k}\right)\right\|^2$$

$$\leq (1-\beta)^2\frac{\sigma^2}{NK}+\beta^2\left(\frac{1}{NK}\sum_{i,k}L^2\left\|\boldsymbol{\theta}^0-\boldsymbol{\theta}_i^{0,k}\right\|^2+\frac{\sigma^2}{NK}\right)$$

$$\leq \frac{\sigma^2}{NK}+\frac{1}{3}\beta^2\eta^2 K^2 L^2.$$

Based on the facts that $(1-\beta) \leq \left(1-\frac{\beta}{2}\right)^2$, $\mathbb{E}\|\boldsymbol{\alpha}\| \leq \sqrt{\mathbb{E}\|\boldsymbol{\alpha}\|^2}$ for any vector $\boldsymbol{\alpha}$, and $\sqrt{a+b+c} \leq \sqrt{a}+\sqrt{b}+\sqrt{c}$ for

any $a, b, c \geq 0$, by taking square root on both side of Equation (3), we have

$$\mathbb{E}\left\|\nabla f\left(\boldsymbol{\theta}^t\right) - \boldsymbol{m}^t\right\| \leq \left(1 - \frac{\beta}{2}\right)^t \left(\frac{\sigma}{\sqrt{NK}} + \beta\eta KL\right) + \left(\frac{2\gamma L}{\beta} + \eta KL + \frac{\sqrt{\beta}\sigma}{\sqrt{NK}}\right),$$

which proves term $i$).

Similarly, following the above process on $\mathbb{E}\left\|\nabla f_i\left(\boldsymbol{\theta}^t\right) - \boldsymbol{m}_i^t\right\|^2$ for each $i$ will give the result of term $ii$). $\qquad\square$

**Lemma 3.** *For any $t$, the error of global descent directions caused by partial client participation is upper bounded by:*

$$\mathbb{E}\left\|\boldsymbol{m}^t - \boldsymbol{g}^t\right\| \leq \frac{2\beta\sigma}{\sqrt{SK}} + \frac{2}{\sqrt{S}}\beta\eta KL + \frac{3}{\sqrt{S}}\gamma L + \frac{\sqrt{3}\beta}{\sqrt{S}}\left(\frac{\sigma}{\sqrt{SK}} + \beta\eta KL\right)\left(1 - \frac{\beta}{2}\right)^t.$$

*Proof.* Define $\boldsymbol{\omega}_i := (1 - \beta)\boldsymbol{m}_i^{t-1} + \beta\boldsymbol{h}_i^t - \boldsymbol{c}_i^{t-1} + \boldsymbol{c}^{t-1} - (1 - \beta)\boldsymbol{m}^{t-1} - \beta\boldsymbol{h}^t$. Then, $\overline{\boldsymbol{\omega}} := \frac{1}{N}\sum_i \boldsymbol{\omega}_i = \mathbf{0}$. Based on the result of Lemma 1, we have

$$
\begin{aligned}
\mathbb{E}\left\|\boldsymbol{m}^t - \boldsymbol{g}^t\right\|^2 &\leq \frac{1}{SN}\sum_i \mathbb{E}\left\|(1 - \beta)\boldsymbol{m}_i^{t-1} + \beta\boldsymbol{h}_i^t - \boldsymbol{c}_i^{t-1} + \boldsymbol{c}^{t-1} - (1 - \beta)\boldsymbol{m}^{t-1} - \beta\boldsymbol{h}^t\right\|^2 \\
&= \frac{1}{SN}\sum_i \mathbb{E}\left\|(1 - \beta)\boldsymbol{m}_i^{t-1} + \beta\boldsymbol{h}_i^t - \boldsymbol{c}_i^{t-1}\right\|^2 - \frac{1}{S}\left\|\boldsymbol{c}^{t-1} - (1 - \beta)\boldsymbol{m}^{t-1} - \beta\boldsymbol{h}^t\right\|^2 \\
&\leq \frac{1}{SN}\sum_i \mathbb{E}\left\|(1 - \beta)\boldsymbol{m}_i^{t-1} + \beta\boldsymbol{h}_i^t - \boldsymbol{c}_i^{t-1}\right\|^2 \\
&= \frac{\beta^2}{SN}\sum_i \mathbb{E}\left\|\boldsymbol{h}_i^t - \boldsymbol{m}_i^{t-1}\right\|^2,
\end{aligned}
$$

where the last equation uses $\boldsymbol{m}_i^{t-1} = \boldsymbol{c}_i^{t-1}$.

$$
\begin{aligned}
\frac{1}{N}\sum_i \mathbb{E}\left\|\boldsymbol{h}_i^t - \boldsymbol{m}_i^{t-1}\right\|^2 &= \frac{1}{N}\sum_i \mathbb{E}\left\|\boldsymbol{h}_i^t \mp \frac{1}{K}\sum_k \nabla f_i\left(\boldsymbol{\theta}_i^{t,k}\right) - \boldsymbol{m}_i^{t-1}\right\|^2 \\
&\leq \frac{\sigma^2}{K} + \frac{1}{N}\sum_i \mathbb{E}\left\|\frac{1}{K}\sum_k \nabla f_i\left(\boldsymbol{\theta}_i^{t,k}\right) \mp \nabla f_i\left(\boldsymbol{\theta}^t\right) \mp \nabla f_i\left(\boldsymbol{\theta}^{t-1}\right) - \boldsymbol{m}_i^{t-1}\right\|^2 \\
&\leq \frac{\sigma^2}{K} + \eta^2 K^2 L^2 + 3\gamma^2 L^2 + \frac{3}{N}\sum_i \mathbb{E}\left\|\nabla f_i\left(\boldsymbol{\theta}^{t-1}\right) - \boldsymbol{m}_i^{t-1}\right\|^2.
\end{aligned}
$$

By Lemma 2, we know that

$$\frac{1}{N}\sum_i \mathbb{E}\left\|\nabla f_i\left(\boldsymbol{\theta}^{t-1}\right) - \boldsymbol{m}_i^{t-1}\right\|^2 \leq (1 - \beta)^t \left(\frac{\sigma^2}{K} + \beta^2\eta^2 K^2 L^2\right) + \left(\frac{2\gamma^2 L^2}{\beta^2} + \eta^2 K^2 L^2 + \frac{\beta\sigma^2}{K}\right).$$

Then, we have

$$
\begin{aligned}
&\frac{1}{N}\sum_i \mathbb{E}\left\|\boldsymbol{h}_i^t - \boldsymbol{m}_i^{t-1}\right\|^2 \\
&\leq \frac{\sigma^2}{K} + \eta^2 K^2 L^2 + 3\gamma^2 L^2 + 3(1 - \beta)^t \left(\frac{\sigma^2}{K} + \beta^2\eta^2 K^2 L^2\right) \\
&\quad + 3\left(\frac{2\gamma^2 L^2}{\beta^2} + \eta^2 K^2 L^2 + \frac{\beta\sigma^2}{K}\right) \\
&= (1 + 3\beta)\frac{\sigma^2}{K} + 4\eta^2 K^2 L^2 + 3\left(1 + \frac{2}{\beta^2}\right)\gamma^2 L^2 + 3(1 - \beta)^t \left(\frac{\sigma^2}{K} + \beta^2\eta^2 K^2 L^2\right)
\end{aligned}
$$

and correspondingly

$$\mathbb{E}\left\|\boldsymbol{m}^t - \boldsymbol{g}^t\right\|^2 \le (1+3\beta)\frac{\beta^2\sigma^2}{SK} + \frac{4}{S}\beta^2\eta^2 K^2 L^2 + \frac{3}{S}\left(\beta^2+2\right)\gamma^2 L^2 + \frac{3}{S}\beta^2(1-\beta)^t\left(\frac{\sigma^2}{K} + \beta^2\eta^2 K^2 L^2\right)$$

$$\le \frac{4\beta^2\sigma^2}{SK} + \frac{4}{S}\beta^2\eta^2 K^2 L^2 + \frac{9}{S}\gamma^2 L^2 + \frac{3\beta^2}{S}\left(\frac{\sigma^2}{SK} + \beta^2\eta^2 K^2 L^2\right)\left(1 - \frac{\beta}{2}\right)^{2t},$$

where the alst inequality uses $(1-\beta) \le \left(1-\frac{\beta}{2}\right)^2$. Since $\mathbb{E}\left\|\boldsymbol{\alpha}\right\| \le \sqrt{\mathbb{E}\left\|\boldsymbol{\alpha}\right\|^2}$ for any vector $\boldsymbol{\alpha}$ and $\sqrt{a+b+c} \le \sqrt{a} + \sqrt{b} + \sqrt{c}$ for any $a, b, c \ge 0$, taking square root on both side gives

$$\mathbb{E}\left\|\boldsymbol{m}^t - \boldsymbol{g}^t\right\| \le \frac{2\beta\sigma}{\sqrt{SK}} + \frac{2}{\sqrt{S}}\beta\eta KL + \frac{3}{\sqrt{S}}\gamma L + \frac{\sqrt{3}\beta}{\sqrt{S}}\left(\frac{\sigma}{\sqrt{SK}} + \beta\eta KL\right)\left(1 - \frac{\beta}{2}\right)^t.$$

$\square$

## C. Proof of Theorem 2

Following the procedures of Equation (1), we know that

$$\frac{1}{T}\sum_t \mathbb{E}\left\|\nabla f\left(\boldsymbol{\theta}^t\right)\right\| \le \frac{\Delta}{\gamma T} + \frac{2}{T}\sum_t \mathbb{E}\left\|\nabla f\left(\boldsymbol{\theta}^t\right) - \widetilde{\boldsymbol{g}}^t\right\| + \frac{\gamma L}{2}.$$

Similarly, define $\boldsymbol{h}_i^t := \frac{1}{K}\sum_k \nabla F\left(\boldsymbol{\theta}_i^{t,k}; \boldsymbol{\xi}_i^{t,k}\right), \boldsymbol{h}^t := \frac{1}{N}\sum_i \boldsymbol{h}_i^t,$

$$\boldsymbol{m}^t := \frac{1}{N}\sum_i \boldsymbol{m}_i^t = \frac{1}{N}\sum_i \left((1-\beta)\boldsymbol{m}_i^{t-1} + \beta\boldsymbol{h}_i^t\right) = (1-\beta)\boldsymbol{m}^{t-1} + \beta\boldsymbol{h}^t,$$

$$\boldsymbol{g}^t := \frac{1}{S}\sum_{i\in\mathcal{S}_t}\boldsymbol{\delta}_i^t + \boldsymbol{c}^{t-1} = \frac{1}{S}\sum_{i\in\mathcal{S}_t}\left((1-\beta)\boldsymbol{m}_i^{t-1} + \beta\boldsymbol{h}_i^t - \boldsymbol{c}_i^{t-1} + \boldsymbol{c}^{t-1}\right).$$

We have $\mathbb{E}[\boldsymbol{m}^t - \boldsymbol{g}^t] = \mathbb{E}\left[\boldsymbol{m}^t - \mathbb{E}_{\mathcal{S}_t}[\boldsymbol{g}^t]\right] = \boldsymbol{0}$. Then, we have

$$\mathbb{E}\left\|\nabla f\left(\boldsymbol{\theta}^t\right) - \widetilde{\boldsymbol{g}}^t\right\|^2 = \mathbb{E}\left\|\nabla f\left(\boldsymbol{\theta}^t\right) - \boldsymbol{m}^t + \boldsymbol{g}^t - \widetilde{\boldsymbol{g}}^t + (\boldsymbol{m}^t - \boldsymbol{g}^t)\right\|^2$$

$$\le \mathbb{E}\left\|\nabla f\left(\boldsymbol{\theta}^t\right) - \boldsymbol{m}^t + \boldsymbol{g}^t - \widetilde{\boldsymbol{g}}^t\right\|^2 + \mathbb{E}\left\|\boldsymbol{m}^t - \boldsymbol{g}^t\right\|^2$$

$$\le 2\mathbb{E}\left\|\nabla f\left(\boldsymbol{\theta}^t\right) - \boldsymbol{m}^t\right\|^2 + 2\mathbb{E}\left\|\boldsymbol{g}^t - \widetilde{\boldsymbol{g}}^t\right\|^2 + \mathbb{E}\left\|\boldsymbol{m}^t - \boldsymbol{g}^t\right\|^2.$$

Taking square root on both sides and then summing over $t$ gives

$$\frac{1}{T}\sum_t \mathbb{E}\left\|\nabla f\left(\boldsymbol{\theta}^t\right) - \widetilde{\boldsymbol{g}}^t\right\| \le \frac{\sqrt{2}}{T}\sum_t \mathbb{E}\left\|\nabla f\left(\boldsymbol{\theta}^t\right) - \boldsymbol{m}^t\right\| + \frac{\sqrt{2}}{T}\sum_t \mathbb{E}\left\|\boldsymbol{g}^t - \widetilde{\boldsymbol{g}}^t\right\|$$

$$+ \frac{1}{T}\sum_t \mathbb{E}\left\|\boldsymbol{m}^t - \boldsymbol{g}^t\right\|. \tag{4}$$

Here, we need to bound an additional term $\frac{\sqrt{2}}{T}\sum_t \mathbb{E}\left\|\boldsymbol{g}^t - \widetilde{\boldsymbol{g}}^t\right\|$ related to the compression error.

First, we have

$$
\mathbb{E}\left\|\boldsymbol{g}^t - \widetilde{\boldsymbol{g}}^t\right\|^2 = \mathbb{E}\left\|\frac{1}{S}\sum_{i \in \mathcal{S}_t}\left(\boldsymbol{\delta}_i^t - \widetilde{\boldsymbol{\delta}}_i^t\right)\right\|^2
$$

$$
\leq \mathbb{E}\left[\frac{1}{S}\sum_{i \in \mathcal{S}_t}\left\|\boldsymbol{\delta}_i^t - \widetilde{\boldsymbol{\delta}}_i^t\right\|^2\right]
$$

$$
= \frac{1}{N}\sum_i \mathbb{E}\left\|\boldsymbol{\delta}_i^t - \widetilde{\boldsymbol{\delta}}_i^t\right\|^2
$$

$$
\leq \frac{1}{N}\sum_i q_i^2 \mathbb{E}\left\|(1-\beta)\boldsymbol{m}_i^{t-1} + \beta \boldsymbol{h}_i^t - \boldsymbol{c}_i^{t-1}\right\|^2,
$$

where the last inequality is based on the $q^2$-contractive compressor assumption. Further, we have

$$
\mathbb{E}\left\|(1-\beta)\boldsymbol{m}_i^{t-1} + \beta \boldsymbol{h}_i^t - \boldsymbol{c}_i^{t-1}\right\|^2
$$

$$
= \mathbb{E}\left\|\boldsymbol{m}_i^{t-1} - \boldsymbol{c}_i^{t-1} + \beta\left(\boldsymbol{h}_i^t \mp \frac{1}{K}\sum_k \nabla f_i\left(\boldsymbol{\theta}_i^{t,k}\right) - \boldsymbol{m}_i^{t-1}\right)\right\|^2
$$

$$
= \mathbb{E}\left\|\boldsymbol{m}_i^{t-1} - \boldsymbol{c}_i^{t-1} + \beta\left(\frac{1}{K}\sum_k \nabla f_i\left(\boldsymbol{\theta}_i^{t,k}\right) \mp \nabla f_i\left(\boldsymbol{\theta}^t\right) \mp \nabla f_i\left(\boldsymbol{\theta}^{t-1}\right) - \boldsymbol{m}_i^{t-1}\right)\right\|^2 + \frac{\beta^2\sigma^2}{K}
$$

$$
\leq 4\mathbb{E}\left(\left\|\boldsymbol{m}_i^{t-1} - \boldsymbol{c}_i^{t-1}\right\|^2 + \frac{1}{3}\beta^2\eta^2 K^2 L^2 + \beta^2\gamma^2 L^2 + \beta^2\left\|\nabla f_i\left(\boldsymbol{\theta}^{t-1}\right) - \boldsymbol{m}_i^{t-1}\right\|^2\right) + \frac{\beta^2\sigma^2}{K}.
$$

From Lemma 2 and Lemma 4, we know that

$$
\mathbb{E}\left\|\nabla f_i\left(\boldsymbol{\theta}^{t-1}\right) - \boldsymbol{m}_i^{t-1}\right\|^2 \leq (1-\beta)^t\left(\frac{\sigma^2}{K} + \beta^2\eta^2 K^2 L^2\right) + \left(\frac{2\gamma^2 L^2}{\beta^2} + \eta^2 K^2 L^2 + \frac{\beta\sigma^2}{K}\right).
$$

$$
\mathbb{E}\left\|\boldsymbol{m}_i^t - \boldsymbol{c}_i^t\right\|^2 \leq \frac{7q_i^2\beta^2\eta^2 K^2 L^2}{(1-q_i)^2} + \frac{9q_i^2\beta^2\gamma^2 L^2}{(1-q_i)^2} + \frac{7q_i^2\beta^2\sigma^2}{K(1-q_i)^2}
$$

Thus, we have

$$
\mathbb{E}\left\|\boldsymbol{g}^t - \widetilde{\boldsymbol{g}}^t\right\|^2
$$

$$
\leq \frac{4}{N}\sum_i q_i^2\left(\frac{7q_i^2\beta^2\eta^2 K^2 L^2}{(1-q_i)^2} + \frac{9q_i^2\beta^2\gamma^2 L^2}{(1-q_i)^2} + \frac{7q_i^2\beta^2\sigma^2}{K(1-q_i)^2} + \frac{1}{3}\beta^2\eta^2 K^2 L^2 + \beta^2\gamma^2 L^2\right.
$$

$$
\left. + \beta^2\left(\left(1-\frac{\beta}{2}\right)^{2t}\left(\frac{\sigma^2}{K} + \beta^2\eta^2 K^2 L^2\right) + \left(\frac{2\gamma^2 L^2}{\beta^2} + \eta^2 K^2 L^2 + \frac{\beta\sigma^2}{K}\right)\right)\right) + \frac{\beta^2\sigma^2}{NK}\sum_i q_i^2.
$$

Since $\sqrt{a+b+c} \leq \sqrt{a} + \sqrt{b} + \sqrt{c}$ for any $a, b, c \geq 0$, taking square root on both side and summing over $t$ gives

$$
\frac{1}{T}\sum_t \mathbb{E}\left\|\boldsymbol{g}^t - \widetilde{\boldsymbol{g}}^t\right\| \leq \sqrt{\frac{1}{N}\sum_i\frac{q_i^4}{(1-q_i)^2}}\left(2\sqrt{7}\beta\eta KL + 6\beta\gamma L + \frac{2\sqrt{7}\beta\sigma}{\sqrt{K}}\right)
$$

$$
+ 2\sqrt{\frac{1}{N}\sum_i q_i^2}\left(\left(\frac{2}{\sqrt{3}} + \frac{2}{T}\right)\beta\eta KL + \beta\gamma L + \sqrt{2}\gamma L + \frac{2\sigma}{T\sqrt{K}} + \frac{\beta(\sqrt{\beta}+1)\sigma}{\sqrt{K}}\right).
$$

Let $\beta = \frac{\sqrt{SK}}{\sqrt{T}}$, $\eta = \frac{1}{K(SKT)^{1/4}}$, and $\gamma = \frac{(SK)^{1/4}}{T^{3/4}}$. Then, we have

$$\frac{1}{T}\sum_t \mathbb{E}\left\|\boldsymbol{g}^t - \widetilde{\boldsymbol{g}}^t\right\| \leq \mathcal{O}\left(\sqrt{\frac{1}{N}\sum_i \frac{q_i^4}{(1-q_i)^2}}\left(\frac{(SK)^{1/4}L}{T^{3/4}} + \frac{\sqrt{S}\sigma}{\sqrt{T}}\right)\right.$$
$$\left.+ \sqrt{\frac{1}{N}\sum_i q_i^2}\left(\frac{(SK)^{1/4}L}{T^{3/4}} + \frac{\sqrt{S}\sigma}{\sqrt{T}}\right)\right).$$

From Lemma 2, we know that

$$\frac{1}{T}\sum_t \mathbb{E}\left\|\nabla f\left(\boldsymbol{\theta}^t\right) - \boldsymbol{m}^t\right\| \leq \frac{2}{\beta T}\left(\frac{\sigma}{\sqrt{NK}} + \beta\eta KL\right) + \frac{2\gamma L}{\beta} + \eta KL + \frac{\sqrt{\beta}\sigma}{\sqrt{NK}}$$
$$\leq \mathcal{O}\left(\frac{\sigma}{K\sqrt{SNT}} + \frac{L}{(NKT)^{1/4}} + \frac{S^{1/4}\sigma}{\sqrt{N}(KT)^{1/4}}\right).$$

By Lemma 3, we have

$$\frac{1}{T}\sum_t \mathbb{E}\left\|\boldsymbol{m}^t - \boldsymbol{g}^t\right\| \leq \frac{2\beta\sigma}{\sqrt{SK}} + \frac{2}{\sqrt{S}}\beta\eta KL + \frac{3}{\sqrt{S}}\gamma L + \frac{2\sqrt{3}}{T\sqrt{S}}\left(\frac{\sigma}{\sqrt{SK}} + \beta\eta KL\right)$$
$$\leq \mathcal{O}\left(\frac{\sigma}{\sqrt{T}} + \frac{K^{1/4}L}{S^{1/4}T^{3/4}}\right).$$

Thus,

$$\frac{1}{T}\sum_t \mathbb{E}\left\|\nabla f\left(\boldsymbol{\theta}^t\right)\right\| \leq \mathcal{O}\left(\frac{\Delta + L}{(NKT)^{1/4}} + \left(\sqrt{\frac{1}{N}\sum_i \frac{q_i^4}{(1-q_i)^2}} + \sqrt{\frac{1}{N}\sum_i q_i^2}\right)\frac{\sqrt{S}\sigma}{\sqrt{T}} + \frac{\sigma}{\sqrt{T}}\right).$$

## C.1. Proof of Technical Lemmas

**Lemma 4.** *For any $i$ and $t$, we have*

$$\mathbb{E}\left\|\boldsymbol{m}_i^t - \boldsymbol{c}_i^t\right\|^2 \leq \frac{7q_i^2\beta^2\eta^2 K^2 L^2}{(1-q_i)^2} + \frac{9q_i^2\beta^2\gamma^2 L^2}{(1-q_i)^2} + \frac{7q_i^2\beta^2\sigma^2}{K(1-q_i)^2}.$$

*Proof.* Since for any $t$, the $S$ elements in $\mathcal{S}_t$ are uniformly sampled from $\{1, \cdots, N\}$, we have

$$\left(\boldsymbol{m}_i^t, \boldsymbol{c}_i^t\right) = \begin{cases} \left(\boldsymbol{m}_i^t, \boldsymbol{c}_i^{t-1} + \mathcal{C}\left(\boldsymbol{m}_i^t - \boldsymbol{c}_i^{t-1}\right)\right) & \text{if } i \in \mathcal{S}_t \\ \left(\boldsymbol{m}_i^{t-1}, \boldsymbol{c}_i^{t-1}\right) & \text{otherwise.} \end{cases}$$

$$\mathbb{E}\left\|\boldsymbol{m}_i^t - \boldsymbol{c}_i^t\right\|^2 = \left(1 - \frac{S}{N}\right)\mathbb{E}\left\|\boldsymbol{m}_i^{t-1} - \boldsymbol{c}_i^{t-1}\right\|^2 + \frac{S}{N}\mathbb{E}\left\|\boldsymbol{m}_i^t - \boldsymbol{c}_i^{t-1} - \mathcal{C}\left(\boldsymbol{m}_i^t - \boldsymbol{c}_i^{t-1}\right)\right\|^2$$
$$\leq \left(1 - \frac{S}{N}\right)\mathbb{E}\left\|\boldsymbol{m}_i^{t-1} - \boldsymbol{c}_i^{t-1}\right\|^2 + \frac{Sq_i^2}{N}\mathbb{E}\left\|(1-\beta)\boldsymbol{m}_i^{t-1} + \beta\boldsymbol{h}_i^t - \boldsymbol{c}_i^{t-1}\right\|^2.$$

For the second term, we have

$$\mathbb{E}\left\|(1-\beta)\boldsymbol{m}_i^{t-1} + \beta\boldsymbol{h}_i^t - \boldsymbol{c}_i^{t-1}\right\|^2$$
$$\leq \mathbb{E}\left\|\boldsymbol{m}_i^{t-1} - \boldsymbol{c}_i^{t-1} + \beta\left(\frac{1}{K}\sum_k \nabla f_i\left(\boldsymbol{\theta}_i^{t,k}\right) \mp \nabla f_i\left(\boldsymbol{\theta}^t\right) \mp \nabla f_i\left(\boldsymbol{\theta}^{t-1}\right) - \boldsymbol{m}_i^{t-1}\right)\right\|^2 + \frac{\beta^2\sigma^2}{K}$$
$$\leq \frac{1}{q_i}\mathbb{E}\left\|\boldsymbol{m}_i^{t-1} - \boldsymbol{c}_i^{t-1}\right\|^2 + \frac{\beta^2}{1-q_i}\left(\eta^2 K^2 L^2 + 3\gamma^2 L^2 + 3\left\|\nabla f_i\left(\boldsymbol{\theta}^{t-1}\right) - \boldsymbol{m}_i^{t-1}\right\|^2\right) + \frac{\beta^2\sigma^2}{K},$$

where the last inequality is based on the fact that $\mathbb{E}\|\boldsymbol{a} + \boldsymbol{b}\|^2 \leq \left(1 + \frac{1-q}{q}\right) \mathbb{E}\|\boldsymbol{a}\|^2 + \left(1 + \frac{q}{1-q}\right) \mathbb{E}\|\boldsymbol{b}\|^2$ for any vectors $\boldsymbol{a}$ and $\boldsymbol{b}$ and $q \in [0, 1)$. Thus,

$$
\begin{aligned}
&\mathbb{E}\left\|\boldsymbol{m}_i^t - \boldsymbol{c}_i^t\right\|^2 \\
&\leq \left(1 - \frac{S(1 - q_i)}{N}\right) \mathbb{E}\left\|\boldsymbol{m}_i^{t-1} - \boldsymbol{c}_i^{t-1}\right\|^2 + \frac{3Sq_i^2\beta^2}{N(1 - q_i)}\mathbb{E}\left\|\nabla f_i\left(\boldsymbol{\theta}^{t-1}\right) - \boldsymbol{m}_i^{t-1}\right\|^2 \\
&\quad + \frac{Sq_i^2\beta^2\eta^2K^2L^2}{N(1-q)} + \frac{3Sq_i^2\beta^2\gamma^2L^2}{N(1-q)} + \frac{Sq_i^2\beta^2\sigma^2}{NK} \\
&\overset{(a)}{\leq} \left(1 - \frac{S(1 - q_i)}{N}\right) \mathbb{E}\left\|\boldsymbol{m}_i^{t-1} - \boldsymbol{c}_i^{t-1}\right\|^2 + \frac{Sq_i^2\beta^2\eta^2K^2L^2}{N(1-q_i)} + \frac{3Sq_i^2\beta^2\gamma^2L^2}{N(1-q_i)} + \frac{Sq_i^2\beta^2\sigma^2}{NK} \\
&\quad + \frac{3Sq_i^2\beta^2}{N(1-q_i)}\left((1-\beta)^{t-1}\left(\frac{\sigma^2}{K} + \beta^2\eta^2K^2L^2\right) + \left(\frac{2\gamma^2L^2}{\beta^2} + \eta^2K^2L^2 + \frac{\beta\sigma^2}{K}\right)\right) \\
&\overset{(b)}{\leq} \sum_{\tau=0}^{t-1}\left(1 - \frac{S(1 - q_i)}{N}\right)^{\tau}\left(\frac{Sq_i^2\beta^2\eta^2K^2L^2}{N(1-q_i)} + \frac{3Sq_i^2\beta^2\gamma^2L^2}{N(1-q_i)} + \frac{Sq_i^2\beta^2\sigma^2}{NK}\right. \\
&\quad + \left.\frac{3Sq_i^2\beta^2}{N(1-q_i)}\left(\frac{2\sigma^2}{K} + \frac{2\gamma^2L^2}{\beta^2} + 2\eta^2K^2L^2\right)\right) \\
&\leq \frac{q_i^2\beta^2\eta^2K^2L^2}{(1-q_i)^2} + \frac{3q_i^2\beta^2\gamma^2L^2}{(1-q_i)^2} + \frac{q_i^2\beta^2\sigma^2}{K(1-q_i)} + \frac{3q_i^2\beta^2}{(1-q_i)^2}\left(\frac{2\sigma^2}{K} + \frac{2\gamma^2L^2}{\beta^2} + 2\eta^2K^2L^2\right) \\
&\leq \frac{7q_i^2\beta^2\eta^2K^2L^2}{(1-q_i)^2} + \frac{9q_i^2\beta^2\gamma^2L^2}{(1-q_i)^2} + \frac{7q_i^2\beta^2\sigma^2}{K(1-q_i)^2}.
\end{aligned}
$$

where $(a)$ is based on the result of Lemma 2, $(b)$ is due to $\boldsymbol{m}_i^{-1} = \boldsymbol{c}_i^{-1}$, $\beta \leq 1$, and $1 - \beta \leq 1$ and $(b)$ is by taking $\beta \leq 1$ and $1 - \beta \leq 1$. □

# D. Additional Numerical Results

## D.1. Simulation Setup

**Tasks and Networks.** We evaluate our algorithms on two widely-used image classification benchmarks: MNIST (LeCun & Cortes, 2005) and Fashion-MNIST (Xiao et al., 2017). MNIST comprises 60,000 training and 10,000 test samples of $28 \times 28$ grayscale handwritten digits $(0 - 9)$. Fashion-MNIST maintains identical dimensions and data splits but contains images of fashion products (e.g., clothing and accessories). For both tasks, we employ a convolutional neural network (CNN) architecture with three convolutional layers and two fully connected layers, yielding $d = 21,840$ trainable parameters.

**Federated Setting.** Our experiments consider a FL system with $N = 100$ clients cooperating to train a shared model. At each global communication round, a fraction of $0.1$ clients are randomly selected to participate, resulting in $S = 10$ active clients per round. Each selected client trains locally for 2 epochs with a batch size of 100 samples. These settings are consistent across all experiments unless otherwise stated.

**Data Heterogeneity.** We evaluate algorithmic robustness under both independent and identically distributed (i.i.d.) and non-i.i.d. data settings. In the i.i.d. setting, data samples are uniformly distributed across clients. For the non-i.i.d. setting, we simulate data heterogeneity using a Dirichlet distribution with parameter $\alpha = 0.1$, where smaller $\alpha$ values indicate greater heterogeneity across client distributions.

**Compressor.** We employ Top-$k$ sparsification with compression ratios $k \in 0.05, 0.1$, applied independently to each network layer. For layer $i$ with parameter dimension $d_i$, the compressor retains $\max(1, \lfloor kd_i \rfloor)$ components with the largest absolute values, setting all remaining components to zero. The maximum operator ensures at least one parameter update per layer, preventing layer stagnation in highly compressed regimes.

**Communication Bits.** We quantify communication overhead assuming 32-bit floating-point encoding for all transmitted parameters. Our analysis considers both uplink and downlink communication, as the cost of downlink broadcasts are typically considered as negligible in federated deployments. Per communication round, the uplink transmission requirements

are $d$ parameters for (Huang et al., 2024a), $d$ for SCAFCOM, and $2d$ for PAdaMFed (Yan et al., 2024), where $d$ denotes the model dimension. Nevertheless, our approach achieves superior downlink communication efficiency compared to both baselines, broadcasting only $d$ parameters while both SCAFCOM and PAdaMFed require $2d$ parameters for server-to-client transmission. All communication costs are reported as average bits per client.

### D.2. Additional Results

Figures 6-8 present training loss trajectories for all evaluated algorithms. The convergence patterns mirror the test accuracy results shown in Figures 1-2, confirming consistent performance across both training and evaluation metrics. For brevity, we omit detailed analysis as the conclusions remain unchanged.

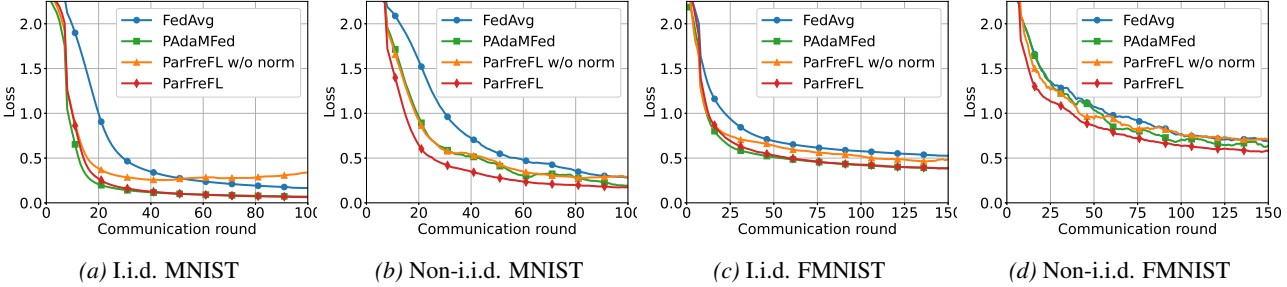

*(a)* I.i.d. MNIST      *(b)* Non-i.i.d. MNIST      *(c)* I.i.d. FMNIST      *(d)* Non-i.i.d. FMNIST

*Figure 6.* Test loss versus communication round of ParFreFL (Algorithm 1) on different datasets with i.i.d./non-i.i.d. data.

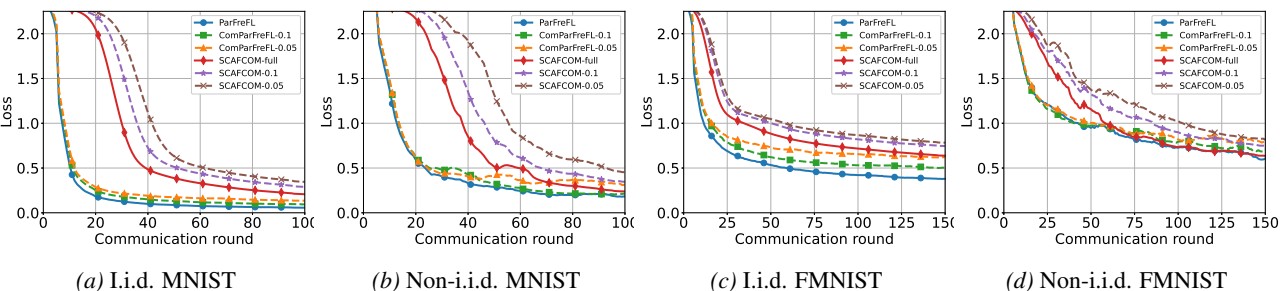

*(a)* I.i.d. MNIST      *(b)* Non-i.i.d. MNIST      *(c)* I.i.d. FMNIST      *(d)* Non-i.i.d. FMNIST

*Figure 7.* Test loss versus communication round of ComParFreFL (Algorithm 2) on different datasets with i.i.d./non-i.i.d. data.

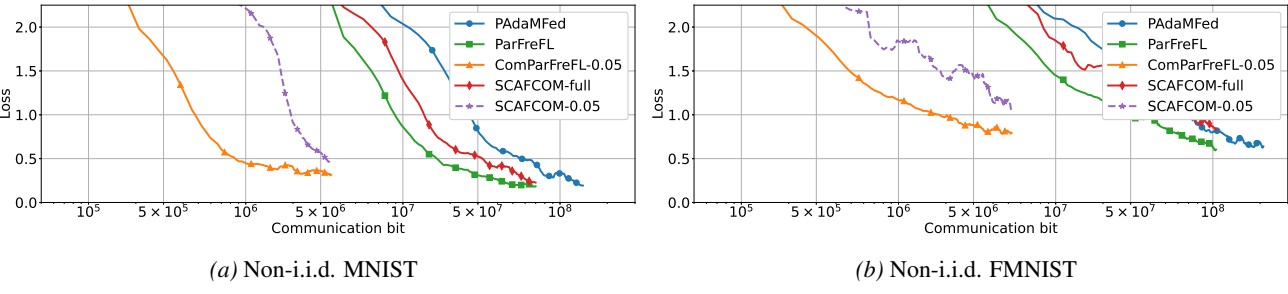

*(a)* Non-i.i.d. MNIST          *(b)* Non-i.i.d. FMNIST

*Figure 8.* Test accuracy versus communication bits of ParFreFL and ComParFreFL on different datasets.

Figure 9 shows the test accuracy versus learning rate for ComParFreFLwith different compression ratios after convergence.

Figure 10 validates ComParFreFLunder dynamic compression ratios between 0.05 and 0.1, indicating the potential efficiency enhancement via dynamically adjusting the compression ratio in our framework.

Figure 11 further compares performance on the CIFAR-10 dataset using ResNet-18, thereby validating the advantage of our methods in more complex federated settings.

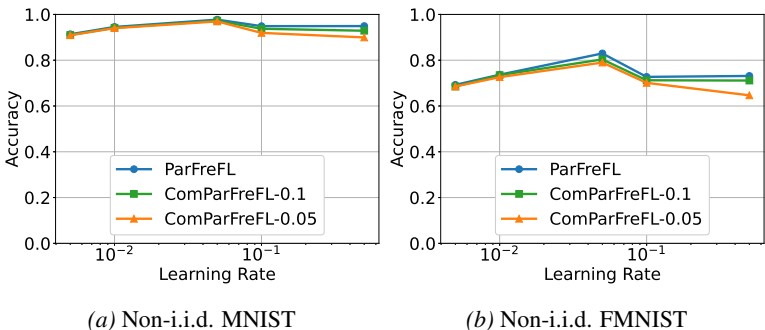

*(a)* Non-i.i.d. MNIST      *(b)* Non-i.i.d. FMNIST

*Figure 9.* Test accuracy versus learning rate under different compression ratios.

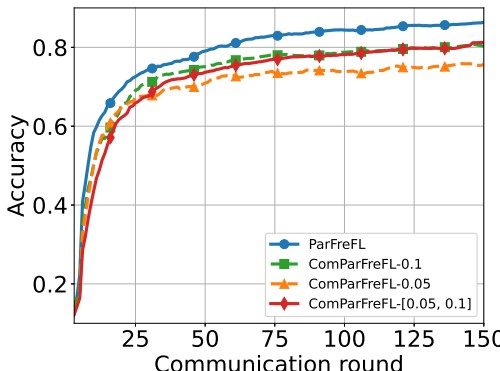

*Figure 10.* Test accuracy versus communication round under dynamic compression ratios (FMNIST).

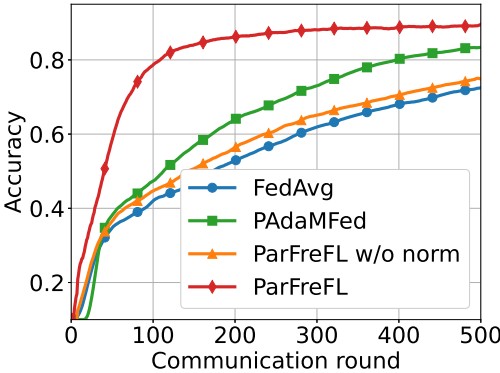

*Figure 11.* Test accuracy versus communication round on CIFAR-10.

