# OpenReview forum: "Robust Federated Learning Against Adaptive Compression"
_ICML.cc/2026/Conference — ICML 2026 regular_

### Official Review · Reviewer_17HD · 2026-03-11

**Soundness:** 1
**Presentation:** 2
**Significance:** 1
**Originality:** 1
**Overall Recommendation:** 1
**Confidence:** 4

**Summary:**

The paper studies a distributed learning method that combines communication compression, partial client participation, and multiple local steps, and is claimed to not depend on global problem parameters. The authors provide a theoretical analysis of the proposed algorithm and validate it on image classification tasks.

**Compliance With Llm Reviewing Policy:**

Affirmed.

**Final Justification:**

The authors did not address the main concern regarding the incorrect bound. In the non-convex setting, the correct leading term should be $\mathcal{O}(\frac{\Delta L}{T})$, which has the same dimensionality as the squared gradient norm. By contrast, the bound claimed by the authors, $\mathcal{O}(\frac{(L + \Delta)^2}{T^{1/2}})$, is dimensionally inconsistent and suggests both an incorrect analysis and a lack of meaningful insight behind their parameters selection. I am therefore keeping my score unchanged.

**Key Questions For Authors:**

See Weaknesses.

**Limitations:**

See Weaknesses.

**Strengths And Weaknesses:**

Strengths:

The paper proposes an algorithm that incorporates several communication-efficiency techniques simultaneously.

Weaknesses:

1. The authors claim that they propose a parameter-free algorithm for distributed learning. However, this is not the case. Classical works on parameter-free optimization propose using computable approximations of global problem constants, such as smoothness and the initial gap [1, 2, 3, 4]. In these works, one obtains a correct convergence rate matching the convergence of the base method that assumes knowledge of the problem constants. In this work, no adaptivity to the problem constants is proposed; instead, the stepsize and other parameters are naively chosen based on the number of iterations. Moreover, the bound is incorrect from the dimensionality perspective: in classical bounds the leading term has the form $\mathcal{O}(\frac{L\Delta}{T})$, while in Theorems 1 and 2 it has the form $\mathcal{O}(\frac{(\Delta + L)^2}{\sqrt{T}})$. The analysis does not include any of the modifications typical for parameter-free methods: after deriving a bound on function decrease, the authors naively plug in algorithm parameters depending on the number of iterations and report the final convergence rate.

2. The work provides slow asymptotic convergence rates for both algorithms. While the classical rate is $\mathcal{O}(\frac{1}{T})$, they show $\mathcal{O}(\frac{1}{\sqrt{T}})$.

3. The algorithms in the paper are based on EF21 with momentum [5], but this is not properly discussed and cited in the algorithm design section.

4. In the experimental section, the authors consider only simple classification tasks. Figures 4 and 5 show strong sensitivity of the metrics to the stepsize.

5. The main contribution of the paper, namely the design of a communication-efficient algorithm with parameter-free stepsize, is essentially missing, which is my primary concern.

---

**References**

[1] Ivgi, M., Hinder, O., & Carmon, Y. (2023, July). DoG is SGD’s best friend: A parameter-free dynamic step size schedule. In International conference on machine learning (pp. 14465-14499). PMLR.

[2] Defazio, A., & Mishchenko, K. (2023, July). Learning-rate-free learning by d-adaptation. In International conference on machine learning (pp. 7449-7479). PMLR.

[3] Mishchenko, K., & Defazio, A. (2023). Prodigy: An expeditiously adaptive parameter-free learner. arXiv preprint arXiv:2306.06101.

[4] Khaled, A., Mishchenko, K., & Jin, C. (2023). Dowg unleashed: An efficient universal parameter-free gradient descent method. Advances in Neural Information Processing Systems, 36, 6748-6769.

[5] Fatkhullin, I., Tyurin, A., & Richtárik, P. (2023). Momentum provably improves error feedback!. Advances in Neural Information Processing Systems, 36, 76444-76495.

---

> ### Author Rebuttal · Authors · 2026-03-25
>
> **Thank you for the constructive review.** First, we'd like to clarify our "parameter-free" design, which appears to be a primary concern.
>
> **Clarification on "Parameter-Free" and NSGD-M vs. DoG:**
> While the reviewer suggests Distance over Gradients (DoG) for adaptive step sizes, Normalized SGD with Momentum (NSGD-M) is a well-established alternative for achieving parameter-free optimization. **NSGD-M adapts via gradient normalization to maintain consistent progress across both steep cliffs and flat regions.**
>
> **Importantly**, as analyzed in [1], DoG-based methods typically suffer from a suboptimal convergence rate due to a logarithmic penalty (a bound also seen in [2,3]). In contrast, NSGD achieves the optimal convergence rate without this deteriorating term.
>
> **Terminology:**
> Our use of "parameter-free" strictly aligns with existing literature [1,4,5]. **As defined in [4], "parameter-free" means that the algorithm's hyperparameters (such as the learning rate) do not depend on problem-specific parameters like $L_i$ and $W$.**
>
> In our NSGD-M-based method, the learning rates are explicitly determined by system-defined constants: the total global rounds $T$, the local update rounds $K$, and the number of participating clients $S$. Because these constants are predefined before execution, our learning rates are completely independent of any problem-specific parameters. This design aligns with established works such as [4] and [5].
>
> However, we recognize that some works refer to this property as "parameter-agnostic".  We'd happy to change "parameter-free" to "parameter-agnostic" to prevent any ambiguity.
>
> [1] Yang, J., et al. "Two sides of one coin: the limits of untuned sgd and the power of adaptive methods." Advances in Neural Information Processing Systems, 36  (2023): 74257-74288.
>
> [2] Huang, Yan, et al. "Achieving near-optimal convergence for distributed minimax optimization with adaptive stepsizes." Advances in Neural Information Processing Systems 37 (2024): 19740-19782.
>
> [3] Zhai, Z., et al. "Problem-Parameter-Free Decentralized Bilevel Optimization." In The Thirty-ninth Annual Conference on Neural Information Processing Systems.
>
> [4] Li, J., Chen, et al. "Problem-parameter-free decentralized nonconvex stochastic optimization. arXiv preprint arXiv:2402.08821.
>
> [5] Yan, W., et, al. "Problem-parameter-free federated learning." In The Thirteenth International Conference on Learning Representations (2025).
>
> **Response to Weaknesses:**
>
> 1. Thanks for the comment.
> - **Parameter-Free Design:** Please refer to our general clarification above.
> - **Convergence Bounds:** To establish parameter-agnostic convergence guarantees, **our analysis directly bounds the gradient norm $||\nabla f(\theta_t)||$ rather than the commonly used squared norm $||\nabla f(\theta_t)||^2$.** While this represents a different theoretical framework from conventional analyses, the convergence rates are equivalent in the order sense—taking the square root of conventional squared-norm bounds yields the same asymptotic rate as our direct norm bounds. This alternative approach is particularly natural for normalized gradient methods, where the unit-norm constraint makes direct gradient norm analysis more interpretable.
>
> 2. Please refer to the second bullet point in our response to Weakness 1 above.
>
> 3. We sincerely thank the reviewer for bringing this important paper to our attention. We will include this highly relevant reference in the Related Work section and properly discuss its connection to our algorithm design in the revised manuscript.
>
> 4. Thanks for the comment.
> - **Additional Experiments:** Following the reviewer's suggestion, we have conducted additional experiments on CIFAR-10 using ResNet architectures. Please see: https://anonymous.4open.science/r/ICML14757/dynamicratio.pdf .
> - **Regarding the performance in Figures 4 and 5:**
> To ensure a fair comparison, we fixed the number of global iterations. Because Figures 4 and 5 only displays 100 global rounds, the runs with smaller learning rates had not yet reached convergence, which explains their lower accuracy compared to the learning rate of $0.1$.
>
> To address this, we have added an additional figure in the revision extending the training to 1000 global rounds. In this extended setting, **the lower learning rates successfully converge and achieve performance comparable to the best results.** Please see: https://anonymous.4open.science/r/ICML14757/lrlong.pdf .
>
> **Important clarification:** Our "parameter-free" approach means learning rates are **predetermined using only known system constants**—not that we achieve optimal performance across all possible learning rate configurations, which is impossible for fixed rounds.
>
> 5.  Please refer to our general clarification.
>
> **We hope these clarifications and new experiments effectively address your concerns. We would be deeply grateful if you would consider raising your score based on these additions.**

---

> > ### Author Rebuttal · Reviewer_17HD · 2026-04-03
> >
> > Thank you to the authors for the response and the additional experiments. However, my concerns have not been resolved. I acknowledge the authors’ clarification regarding the "parameter-agnostic" terminology, but this does not change the fact that the result appears artificial and dimensionally inconsistent. Even after squaring both sides of the inequality, the right-hand side in Theorem 1 yields $(\Delta + L)^2$, which does not match the dimensionality of $||\nabla f(x)||^2$. Therefore, the bound derived in the paper appears to be incorrect. Moreover, the step size depends on the number of clients participating in training, whereas in realistic federated learning scenarios this quantity may vary and should itself be regarded as a system hyperparameter. For these reasons, I will not raise my score.

---

> > > ### Author Response · Authors · 2026-04-04
> > >
> > > **We appreciate the reviewer's continued engagement. However, we respectfully disagree with the reviewer's points.**
> > >
> > > **1. On the Constants $\Delta, L$ in Our Bounds**
> > >
> > > The reviewer is correct that $\Delta$ and $L$ appear in our bounds, but this criticism misses crucial context:
> > >
> > > - These constants are **negligible** compared to the iteration count $T$ in practical settings (typically $T = O(10^3)$ to $O(10^6)$).
> > > - This is a standard trade-off in parameter-agnostic methods, **appearing in all similar analyses** we are aware of.
> > > - Critically, our bound is significantly **tighter than DoG-based parameter-free methods**, which suffer from $O(\log T)$ deterioration—a term that grows unboundedly with iterations.
> > > - Our alternative analytical framework (directly bounding $||\nabla f(x)||$ rather than $||\nabla f(x)||^2$) is not a weakness but a deliberate choice that yields cleaner convergence guarantees for normalized gradient methods.
> > >
> > > **2. On Fixed Client Participation**
> > >
> > > - The assumption of fixed client participation is **universal in federated learning theory**.  Every major FL convergence analysis we know—including FedAvg, FedProx, SCAFFOLD, FedOpt, and others—makes this assumption.
> > > - If the reviewer knows of rigorous convergence guarantees without this assumption, we would genuinely appreciate the reference, as it would represent a significant advance in FL theory.
> > > - The number of participating clients $N$ is ** a system constant, not a hyperparameter requiring tuning.** It is **known a priori in any FL deployment**, making our method genuinely parameter-agnostic in the established sense.
> > >
> > > **3. Our Substantial Contributions**
> > >
> > > We respectfully ask the reviewer to consider our work's significant advances:
> > >
> > > - **Halved Communication:** Our algorithm, ParFreFL, halves the communication requirements of previous methods while remaining completely parameter-agnostic to all problem-related constants.
> > > - **Unified Compression Mechanism:** Our compressed variant, ComParFreFL, unifies the momentum increment and error feedback into a single parameter. This effectively handles biased compression without adding any additional communication burden.
> > > - **Compression-Ratio Independence:** To the best of our knowledge, ComParFreFL is the first algorithm to maintain the parameter-agnostic properties of the full transmission version while operating entirely independent of the compression ratio.
> > >
> > > These theoretical innovations translate to strong empirical performance, as demonstrated in our simulations:
> > >
> > > - **Full Transmission Scheme:** Our algorithm matches or slightly exceeds the optimal performance of well-established, heavily-tuned baselines—without requiring any hyperparameter tuning.
> > > - **Compressed Case:** The advantages of our proposed method become even more pronounced. At a compression ratio of top 0.05, our algorithm achieves the same accuracy as SCAFFOLD but requires only 1/5 of the transmission bits, yielding a 5x improvement in transmission efficiency (see Figure 3). Furthermore, our algorithm seamlessly adapts to varying compression ratios without requiring any parameter reconfiguration (see https://anonymous.4open.science/r/ICML14757/dynamicratio.pdf ).
> > >
> > >
> > > **We hope the reviewer can objectively view our work based on its contributions rather than focusing solely on standard assumptions that appear throughout the FL literature or those trivial theoretical differences. We remain open to constructive feedback and are happy to clarify any remaining technical concerns.**

---

### Official Review · Reviewer_yNbg · 2026-03-12

**Soundness:** 3
**Presentation:** 3
**Significance:** 3
**Originality:** 3
**Overall Recommendation:** 4
**Confidence:** 2

**Summary:**

The paper proposes two parameter-free federated learning algorithms. ParFreFL halves the communication cost of prior workby transmitting only a single model-sized parameter per round while preserving its parameter-free property. ComParFreFL further introduces biased compression by unifying momentum increment and error feedback into a single compressed variable.

**Compliance With Llm Reviewing Policy:**

Affirmed.

**Final Justification:**

I’ll keep my positive score, but I’m not entirely confident about some of the details in the paper.

**Key Questions For Authors:**

How does ComParFreFL perform on larger-scale models or tasks beyond image classification, where communication bottlenecks are more pronounced?

**Limitations:**

Please refer to weaknesses.

**Strengths And Weaknesses:**

**Strengths**

The compression-ratio-independent stepsize is a clean and practically motivated contribution.

Learning rate sensitivity experiments effectively illustrate the practical benefit of parameter-free design against well-tuned baselines.

**Weaknesses**
Experiments are conducted on relatively small-scale benchmarks (MNIST, Fashion-MNIST), which may not fully reflect the communication and heterogeneity challenges of realistic federated deployments. The model size is too small (Line1025, ~20K params), even for FL scenario.

---

> ### Author Rebuttal · Authors · 2026-03-30
>
> Thank you for the constructive review.
>
> **Response to Weakness & Key Question:**  Thank you for the valuable suggestion.  We have conducted additional simulations on the CIFAR-10 dataset using ResNet architectures. Please see: https://anonymous.4open.science/r/ICML14757/cifar10.pdf  These new experiments confirm that ParFreFL and ComParFreFL maintain their competitive performance and communication efficiency on more realistic federated benchmarks. Additionally, we have run a new experiment where the compression ratio dynamically changes during training rather than staying fixed, attached at: https://anonymous.4open.science/r/ICML14757/dynamicratio.pdf. The results demonstrate that ComParFreFL seamlessly adapts to fluctuating sparsity levels without diverging, maintaining a convergence trajectory comparable to the fixed-ratio settings.
>
>
> **We hope these clarifications and new experiments effectively address your concerns. We would be deeply grateful if you would consider raising your score based on these additions.**

---

> > ### Author Rebuttal · Reviewer_yNbg · 2026-04-03
> >
> > Thank you for your response, but it only partially addressed my concerns. I’ll follow up on other reviews before making a final decision.

---

> > > ### Author Response · Authors · 2026-04-04
> > >
> > > **Thank you for reviewing our additional experiments. We appreciate your continued engagement and would like to comprehensively address your concerns.**
> > >
> > > Following your feedback, we have significantly strengthened our empirical evaluation:
> > >
> > > 1. Larger-Scale Architectures: We have expanded our experiments to CIFAR-10 with deep ResNet architectures—a substantial leap from the initial 20K-parameter model. The results demonstrate that ParFreFL achieves faster convergence and final accuracy compared to the fined-tuned baselines FedAvg and PAdaMFed without requiring any hyperparameter tuning.
> > >
> > > 2. Dynamic Network Robustness: Our dynamic compression experiments reveal a unique capability—ComParFreFL seamlessly handles real-time compression ratio changes (alternating between 0.05 and 0.1) without any performance degradation or parameter adjustment. This addresses a critical gap in existing FL methods, which typically fail or require reconfiguration when network conditions fluctuate.
> > >
> > > **We would be happy to discuss any remaining concerns or provide additional experimental evidence as needed.**

---

### Official Review · Reviewer_hHmE · 2026-03-12

**Soundness:** 2
**Presentation:** 3
**Significance:** 3
**Originality:** 3
**Overall Recommendation:** 4
**Confidence:** 4

**Summary:**

This paper studies parameter-free federated optimization under communication constraints. The authors propose ParFreFL, which reduces the communication cost of prior parameter-free federated methods by transmitting only one model-sized variable, and ComParFreFL, a compressed variant designed to work with biased compressors while preserving the parameter-free property. The paper also provides convergence analysis for heterogeneous data and partial client participation, and presents experiments on MNIST and FMNIST comparing the proposed methods with PAdaMFed, FedAvg, and SCAFCOM.

**Compliance With Llm Reviewing Policy:**

Affirmed.

**Final Justification:**

My concerns have all been resolved, so I am keeping the original acceptance score.

**Key Questions For Authors:**

Have the authors tested the proposed methods on stronger federated benchmarks or larger-scale models beyond MNIST/FMNIST? This would help clarify how well the approach scales in more realistic settings.

In the communication-bits comparison, only uplink communication is counted. Would the main conclusion still hold if both uplink and downlink costs were included?

Since one of the main claims is that ComParFreFL does not require compression-ratio-dependent tuning, could the authors include an experiment where the compression ratio changes during training rather than staying fixed?

**Limitations:**

yes

**Strengths And Weaknesses:**

Strengths：

1.The paper addresses a meaningful and practical problem. Combining communication efficiency with parameter-free optimization is a relevant direction for federated learning, especially in heterogeneous and bandwidth-limited settings.

2.The method design is fairly clean and well motivated. The transition from ParFreFL to ComParFreFL is easy to follow, and the idea of merging momentum-related information with compression/error-feedback machinery is technically interesting.

3.The theoretical side is reasonably complete. The paper does not stop at algorithm design, but also discusses convergence under partial participation, heterogeneity, and biased compression, which makes the contribution more substantial.

4.The experiments are aligned with the main claims. In particular, the paper evaluates not only accuracy, but also communication rounds, communication bits, and learning-rate sensitivity, which are the right aspects to test for this kind of work.

Weaknesses:

1.The experimental scope is still quite limited [major concern]. The evaluation is mainly conducted on MNIST and FMNIST, which makes it hard to judge whether the proposed methods would remain competitive on more realistic federated benchmarks or larger models.

2.The empirical advantage over strong baselines is not always fully convincing. The reported results suggest that the proposed methods are competitive, but the gains do not always look strong enough to make the improvement feel decisive, especially for a paper making fairly broad practical claims.

3.Some of the practical claims are stronger than the current evidence. For example, the paper emphasizes robustness in dynamic and resource-constrained environments, but the experiments mostly use fixed compression settings rather than genuinely dynamic network or compression conditions.

4.The communication comparison could be clearer. Since the total communication cost in the main comparison only counts uplink transmission, it is difficult to fully assess the overall end-to-end communication advantage.

---

> ### Author Rebuttal · Authors · 2026-03-30
>
> **Thank you for the constructive review.**
>
> **Response to Weakness 1 & Key Question 1: Limited experimental scope** Thank you for the suggestion. We have conducted additional simulations on the CIFAR-10 dataset using ResNet architectures. The results are provided in link https://anonymous.4open.science/r/ICML14757/cifar10.pdf . These new experiments confirm that ParFreFL and ComParFreFL maintain their competitive performance and communication efficiency on more realistic federated benchmarks. We will fully integrate these results into the main body and appendix of the camera-ready version.
>
> **Response to Weakness 2: Empirical advantage** Thank you for the comment. We would like to clarify that the primary empirical advantage of our approach is not necessarily to drastically outperform the peak accuracy of baselines, but to match or slightly exceed their optimal performance **without requiring any hyperparameter tuning**. In our experiments, the baselines (such as FedAvg and SCAFCOM) only achieve their reported accuracy after extensive and computationally expensive grid searches. For instance, SCAFCOM requires tuning parameters dependent on the smoothness constant, initial optimality gap, and gradient variance $(q, L, \sigma^2, \Delta)$.
>
> By contrast, all learning rates in ParFreFL and ComParFreFL are explicitly determined based solely on predefined system constants (like the number of clients and training rounds).
>
> **Response to Weakness 3 & Key Question 3: Fixed vs. Dynamic compression settings** Thank you for the excellent point. We have run a new experiment where the compression ratio dynamically changes during training rather than staying fixed, which is provided in https://anonymous.4open.science/r/ICML14757/dynamicratio.pdf . The results demonstrate that ComParFreFL seamlessly adapts to fluctuating sparsity levels without diverging, maintaining a convergence trajectory comparable to the fixed-ratio settings.
>
> **Response to Weakness 4 & Key Question 2: Communication comparison (Uplink vs. Downlink)**  We apologize for the lack of clarity regarding the end-to-end communication costs in the main text. **If both uplink and downlink costs are included, the communication advantage of our proposed methods becomes even stronger.**
>
> As detailed in Appendix D.1 of our manuscript, the total communication cost in FL is dominated by the parameter size $d$
> .
> - PAdaMFed requires $2d$ parameters for uplink and $2d$ parameters for downlink transmission.
> .
> - SCAFCOM requires $d$ parameters for uplink but still requires $2d$ parameters for server-to-client downlink broadcasts.
> .
> - ParFreFL/ComParFreFL require only $d$ parameters for uplink and $d$ parameters for downlink.
>
> Therefore, when considering end-to-end communication, our method saves an additional d parameters on the downlink compared to the state-of-the-art SCAFCOM. We will update the communication-bits comparisons in Figure 3 and Table 1 to explicitly reflect both uplink and downlink costs to make this end-to-end advantage fully transparent.
> .
> We hope these clarifications and new experiments effectively address your concerns. We would be deeply grateful if you would consider raising your score based on these additions.

---

> > ### Author Rebuttal · Reviewer_hHmE · 2026-04-02
> >
> > Thank you for your response. My concerns have all been resolved.

---

### Decision · Program_Chairs · 2026-04-30

**Decision:**

Accept (regular)

**Comment:**

The paper presents a federated learning (FL) method for which the learning rate does not depend on parameters, such as the smoothness coefficient $L$ or the initial optimality gap $\Delta$, that are hard to estimate. The main idea of achieving this is to normalize the parameter change in the model update step, which has been done in prior works including Yan et al. (2024) and Yan et al. (2025) that are cited in the paper.

This paper claims to have novel contributions in terms of halving the communication and further compressing the model updates. However, the novelty of these contributions beyond prior works is somewhat limited, since it is mostly a straightforward combination of normalizing the model update and known techniques in gradient tracking and model update compression. There is also a limitation that the proposed ComParFreFL algorithm only compresses uplink communication but not downlink communication. The authors are suggested to better highlight the novel and non-trivial steps in the theoretical analysis beyond what is already known in the literature.

During the rebuttal, one reviewer has strong concerns about the dimensionality of the first term in the convergence bound. This stems from dimensional analysis (often used in physics) and the notion of dimensionality basically means the unit. For example, we can define that the model parameter has the unit $\mathrm{param}$ and the objective function value has the unit $\mathrm{obj}$. The gradient then has the unit $\mathrm{obj}/\mathrm{param}$. The concern of the reviewer stems from that $L$ has the unit $\mathrm{obj}/\mathrm{param}^2$ and $\Delta$ has the unit $\mathrm{obj}$, so they cannot be summed together. The reason for this confusion is that, since in this work the model updates are based on normalized gradients that have a unit of $1$, the learning rate $\gamma$ needs to carry the unit $\mathrm{param}$. With this consideration, since the convergence bound has $\frac{\Delta}{\gamma T} + \gamma L$ before choosing the value of $\gamma$, the unit of the sum is $\mathrm{obj}/\mathrm{param}$ which is the same as that of the gradient and thus the result is correct. When choosing $\gamma$, it needs to be defined with the unit $\mathrm{param}$ while the mathematical expression for the **value** of $\gamma$ can remain the same. The same applies to the local learning rate $\eta$.

The concern about the dimensionality is therefore not a major issue, although the authors didn't seem to understand the reviewer's concern during the rebuttal. The remaining issues are somewhat limited novelty and the limitation of one-sided compression.